# Atmospheric-river-induced foehn events drain glaciers on Novaya Zemlya

J. Haacker [1] ✉, B. Wouters [1] ✉, X. Fettweis [2], I. A. Glissenaar[3,4] & J. E. Box[5]

Recently, climate extremes have been grabbing attention as important drivers of environmental change. Here, we assemble an observational inventory of energy and mass fluxes to quantify the ice loss from glaciers on the Russian High Arctic archipelago of Novaya Zemlya. Satellite altimetry reveals that $70 \pm 19\%$ of the $149 \pm 29$ Gt mass loss between 2011 and 2022 occurred in just four high-melt years. We find that $71 \pm 3\%$ of the melt, including the top melt cases, are driven by extreme energy imports from atmospheric rivers. The majority of ice loss occurs on leeward slopes due to foehn winds. 45 of the 54 high-melt days ($>1$ Gt d$^{-1}$) in 1990 to 2022 show a combination of atmospheric rivers and foehn winds. Therefore, the frequency and intensity of atmospheric rivers demand accurate representation for reliable future glacier melt projections for the Russian High Arctic.

In recent decades, glaciers have demonstrated a pronounced climate sensitivity by losing a larger fraction of their total mass than the ice sheets of Antarctica and Greenland[1,2]. Climate projections indicate an additional reduction of 23 to 43% of their remaining mass by the end of this century[3]. The increased ice loss is to produce an accelerated contribution to sea level rise, affecting the livelihoods of many[4]. Despite this, the confidence in glacier change projections is not very high[5]. These projections are based on the coarsely resolved input of General Circulation Models and, further, mostly use temperature-index based models to estimate the surface melt. Both inhibit the accurate representation of extreme weather events and the associated surface melt, respectively.

The glaciers on Novaya Zemlya have reduced faster than any in the Russian High Arctic Islands region[6]. Enclosed by the Barents Sea with Atlantic influx to its west, and the Kara Sea – characterized by colder fresh surface water – to its east, it contains approximately 7600 km$^3$ [7] of ice distributed among its 479 glaciers, of which 36 terminate into the sea[8]. Novaya Zemlya's glaciers have been in retreat since their maximum extent during the Little Ice Age[9], as meltwater production and runoff increases have outpaced changes in snowfall and rainfall since at least the 1950s, resulting in surface mass budget deficits ranging from 0.9 to 5.2 Gt yr$^{-1}$ [10]. Additionally, to surface losses, the tidewater glaciers now discharge roughly 1 Gt yr$^{-1}$ more ice into the ocean than they did in 2010 to 2020[11].

Ice loss has increased over the past two decades, with a mass loss of $-7.6 \pm 1.2$ Gt yr$^{-1}$ and $-5.8 \pm 3.0$ Gt yr$^{-1}$ observed by spaceborne laser altimetry and gravimetry, respectively, between 2004 and 2009 on Novaya Zemlya[12]. In the period 2010 to 2020, this has increased to $-10.7 \pm 0.9$ Gt yr$^{-1}$, based on CryoSat−2 radar altimetry[6]. Model projections indicate that the glacier volume in Novaya Zemlya may decrease by $27 \pm 9\%$ under Representative Concentration Pathway RCP4.5 by the end of this century[3]. Tepes et al.[13] inferred that an increasing transport of warm subsurface water to the glacier fronts led to a faster glacier flow and elevated mass loss of tidewater glaciers along the Barents coast. However, the discharge estimates of later studies[6,11] do not explain the increase in mass loss despite the observed acceleration of tidewater glaciers[14,15].

Concentrated poleward flows of heat and moisture in atmospheric rivers (ARs) have come into focus as important drivers of melt extremes around Greenland[16–19] and may even trigger calving events at the Antarctic Peninsula[20]. Recently, Ma et al.[21] reported an expected AR frequency increase for 2024 to 2064 compared to 1981 to 2021 for most of the Arctic with especially high rates over the Barents Sea. Here, we detail the role ARs have in ice loss on Novaya Zemlya through a surface energy budget (SEB) and an investigation of atmospheric thermodynamics, vertically integrated water vapor transport, and wind direction.

[1]Department of Geoscience and Remote Sensing, University of Technology Delft, Delft, the Netherlands. [2]Departement of Geography, SPHERES Research Unit, University of Liège, Liège, Belgium. [3]Institute for Marine and Atmospheric Research, Utrecht University, Utrecht, the Netherlands. [4]Bristol Glaciology Centre, School of Geographical Sciences, University of Bristol, Bristol, UK. [5]Department of Glaciology and Climate, Geological Survey of Denmark and Greenland, Copenhagen, Denmark. ✉e-mail: j.m.haacker@tudelft.nl; bert.wouters@tudelft.nl

## Results

### Mass loss over the past decades

In the following, we start at a multi-decadal time scale and proceed to increasingly focus on shorter time scales where we find that melt extremes are overlooked important drivers of glacier change. To estimate the glacier mass changes from 1980 to 2022 following the Input-Output (I/O) method we subtract published ice discharge estimates[11] from the modeled climatic surface mass balance, i.e., snow accumulation minus snow and ice ablation, using the regional atmospheric model MAR (see Methods). The results are visualized in Fig. 1. Combining the changes from all glaciers on Novaya Zemlya, the mass balance of $-5.3 \pm 2.4$ Gt yr$^{-1}$ during 1981 to 2010 is equivalent with the long-term (1952 to 2013/2014) estimate of $-5.1 \pm 0.8$ Gt yr$^{-1}$ based on subtracting digital elevation models[15]. The yearly mass loss increased to $-12.4 \pm 2.3$ Gt yr$^{-1}$ in 2011 to 2022. Of the $7.1 \pm 3.3$ Gt yr$^{-1}$ difference between 1981 to 2010 and 2011 to 2022, $1.7 \pm 1.9$ Gt yr$^{-1}$ result from the modeled increase of ice discharge by tidewater glaciers and $5.5 \pm 2.7$ Gt yr$^{-1}$ are lost via surface processes, mainly meltwater runoff. That the increase of surface losses exceeds that of discharge corroborates the results of Jakob and Gourmelen[6], who linked mass loss on Novaya Zemlya to persistently below-average surface mass balance in this period, and a minor role of mass balance changes due to varying ice flow. Melt, i.e., the daily production of meltwater, increased all over the ice cap in 2011 to 2022 compared to 1981 to 2010. The melt increased more at lower than at higher elevations and it increased most along the Barents coast (see Supplementary Fig. S1).

These model-based I/O results are in line with their observational counterparts from GRACE/GRACE-FO[22], ICESat[12], and CryoSat−2, shown in Fig. 1 (see Methods and Supplementary Table S1). Over the period for which CryoSat−2 observations are available, 2010 to 2022, we find excellent agreement with the model results, namely an average mass loss of $-12.5 \pm 1.6$ Gt yr$^{-1}$. The results are, further, in line with previous studies[6,13,23]. The altimetry data indicate increased thinning rates, but the spatial distribution of elevation trends in 2010 to 2022 generally remained similar to what has been reported for the period 2003 to 2009[12], with most losses at lower elevations. We do, however, find a difference between area-specific change rates of land- and marine-terminating glacier fronts, which were on par in the earlier

period. Between the periods 2003 to 2009 and 2010 to 2022, the mass balance of tidewater glacier fronts (see Methods) decreased from $-900$ kg m$^{-2}$ yr$^{-1}$ to $-1200$ kg m$^{-2}$ yr$^{-1}$. Note that, considering the range 0 to 500 m, tidewater glaciers have larger shares of their surface area at low elevations (see Supplementary Fig. S2) where most melt happens. Furthermore, the region around the ice divide (see Methods) now shows pronounced thinning, and the mass balance became more negative along the Barents Sea than along the Kara Sea. Supplementary Table S2 shows the earlier and present results side-by-side and Supplementary Fig. S3 a change rate map.

### Yearly variability of the mass loss

We find that the 2010 to 2022 mass balance deficit has been highly variable between glaciological years, in this study defined to start on 1 October each year. The standard deviation of yearly mass changes observed with CryoSat−2 is $12 \pm 5$ Gt yr$^{-1}$. Of the total $149 \pm 29$ Gt ice loss in this period, $70 \pm 19\%$ occurred in four years (2013, 2016, 2020, and 2022). From the MAR surface mass balance (SMB) components, we find that the overall variability of $13 \pm 5$ Gt yr$^{-1}$ is with $14 \pm 6$ Gt yr$^{-1}$ mostly caused by the surface melt water production, whereas the snowfall variability contributes a minor part of $4 \pm 2$ Gt yr$^{-1}$. Note, that the melt variability does not propagate completely to the SMB variability because the produced meltwater is partly refrozen and retained by snowpack. The year-to-year variabilities of the rainfall's SMB contribution, i.e., the snowpack-retained rainfall, and the surface sublimation are below 1 Gt yr$^{-1}$. The yearly variability in the model data agrees well with the CryoSat−2 observations, shown by a correlation coefficient of 0.97 ($p = 1 \times 10^{-7}$) and a root-mean-squared difference of 3.0 Gt yr$^{-1}$. Considering the area-specific mass balances of the entire land- and marine-terminating glaciers (see Supplementary Fig. S4), we do not find a termination-type dependence, as we did for the fronts only. The yearly mass balance variability and the termination-type independence indicate that especially the large losses in 2013, 2016, 2020, and 2022 happened mainly via surface melt.

For the period 2011 to 2022, we find that the sensible heat flux, the downward longwave radiation, and the latent heat flux are larger than the 1981 to 2010 MAR based average, while the downward shortwave radiation stays below climatological average (see Fig. 2b). These SEB-

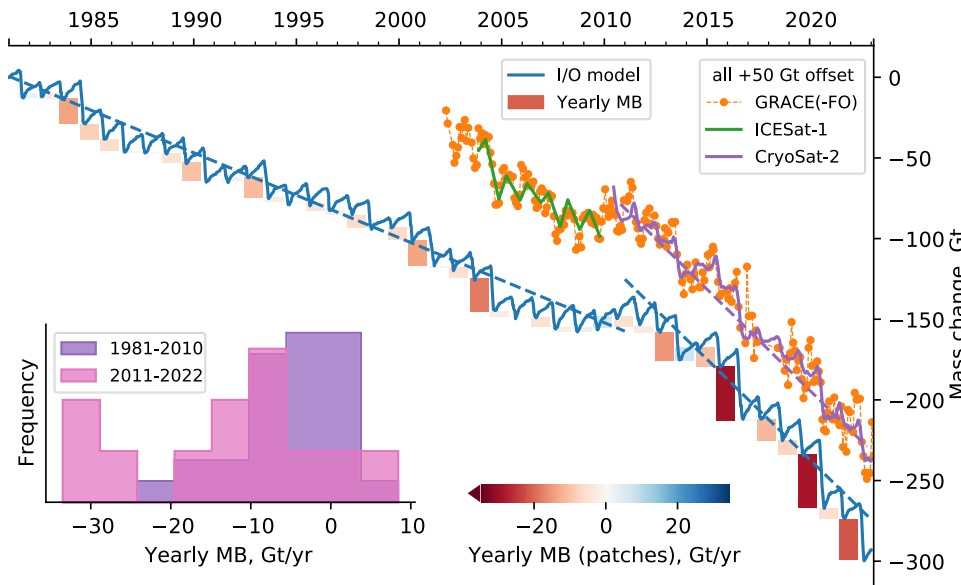

**Fig. 1 | Glacier mass change time series.** The mass change of the ice cap on Novaya Zemlya since 1 January, 1980, derived from the Input-Output (I/O) model. We include observations from GRACE/GRACE-FO[22], ICESat[12], and CryoSat−2 for which we chose arbitrary starting points such that the differences to each other are minimized but retain a 50 Gt-offset to the I/O time series to declutter the plot. The

dashed lines show least-square regression results of the I/O data for the periods 1981 to 2010 and 2011 to 2022 and of the CryoSat−2 data for 2010 to 2022. The colored patches along the I/O time series quantify the prevailing negative glacier mass balance (MB) for glaciological years, i.e., starting 1 October; those are aggregated into the inset histogram. Source data are provided as a Source Data file.

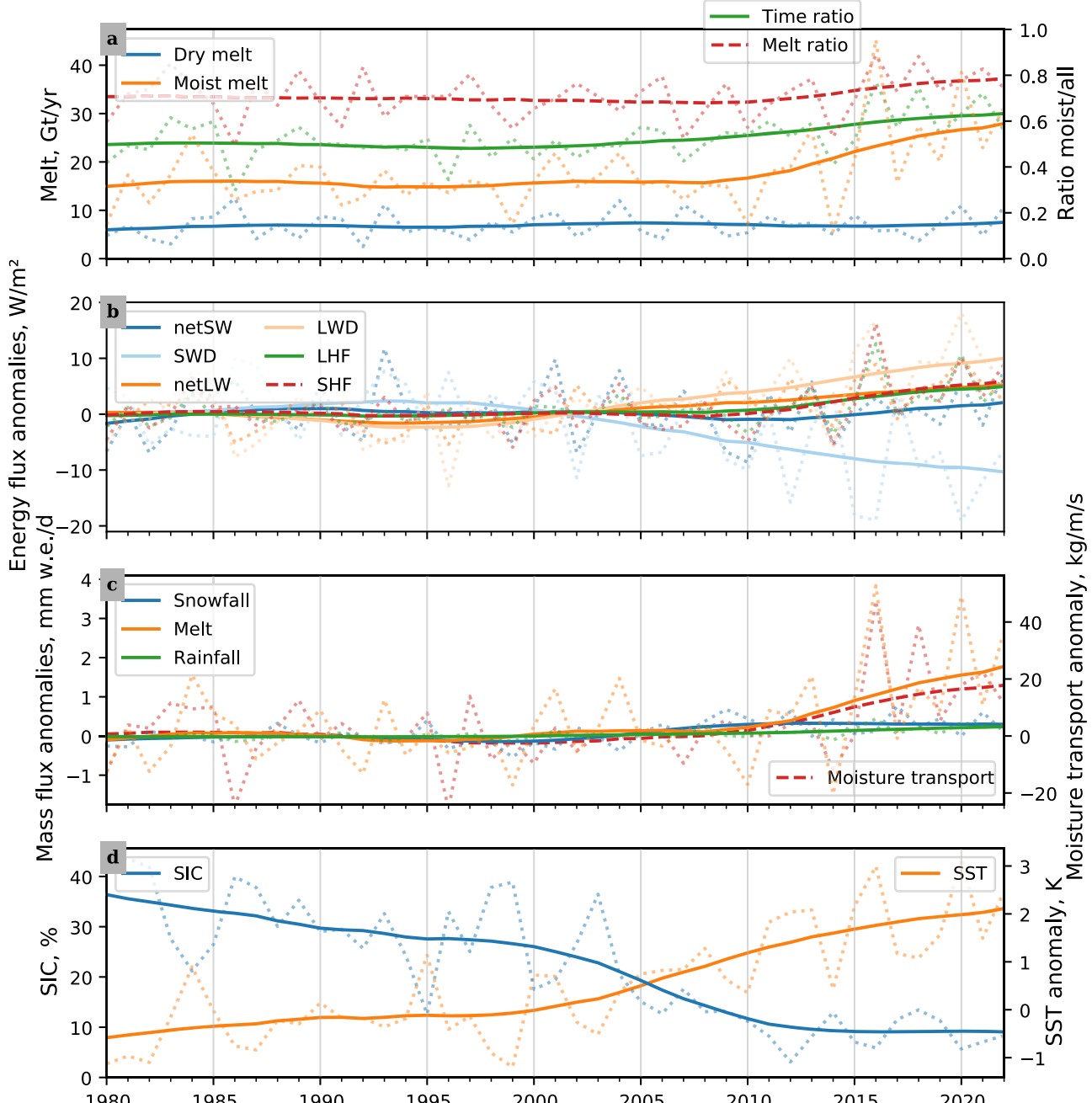

**Fig. 2 | Time series of glacier mass change related variables.** This figure shows spatially aggregated surface mass and energy fluxes and environment variables. In each panel, the variables listed in the left and right legends are associated with the y-axis on the same side. For each time series, the dotted line shows the melt season aggregates, i.e., June until September, and the solid line shows the low-pass filtered version (centered 15 yr moving Gaussian windows with 4 yr standard deviation). **a** shows the regional atmospheric model MAR melt in dry and moist conditions on the left, and the time ratio, i.e., the moist condition's share of days and the melt ratio on the right. The moist time ratio is closely linked to the presence of atmospheric rivers (see Methods and Supplementary Fig. S11). **b** shows the following MAR surface energy budget (SEB) component anomalies (see Methods): net shortwave radiation (netSW), downward shortwave radiation (SWD), net longwave radiation (netLW), downward longwave radiation (LWD), latent heat flux (LHF), and sensible heat flux (SHF). **c** shows the anomalies of the MAR variables snowfall, melt, and rainfall on the left and the Modern-Era Retrospective analysis for Research and Applications, Version 2 (MERRA-2) moisture transport on the right. **d** shows the ECMWF Reanalysis v5 (ERA5) sea ice concentration (SIC) left and the ERA5 sea surface temperature (SST) anomaly right. Source data are provided as a Source Data file.

component anomalies are especially pronounced for the high-melt years. Contrary, in 2014 the SEB anomalies were reversed, contributing to an overall positive mass balance for this year.

We tested whether the variability or the additional mass balance deficit of the recent years was caused by additional melt at the start or end of a melt season, i.e., June to September. We note an insignificant decrease of the June ($p = 0.24$) and September ($p = 0.052$) SMBs in 2011 to 2022 compared to 1981 to 2010 by $-0.6 \pm 1.1$ Gt and $-1.8 \pm 1.8$ Gt,

respectively (see Supplementary Fig. S5). The June- and September-only SMB decreases could neither explain the observed yearly variability, nor the additional longer-term mass balance deficit.

### Importance & evolution of moisture imports

When considering shorter time periods, we find large (up to 1 Gt d$^{-2}$) daily variability in the modeled melt: the melt above its low-pass filtered version (see Methods and Supplementary Figs. S6 and S7)

accounts on average for 36 ± 2% of the yearly totals. The short-term variability does not substantially change throughout 1980 to 2022. We compare the daily melt to a dataset of vertically integrated water vapor transport[18], hereafter moisture transport. Over the model period, on average 71 ± 3% of the melt is committed in moist conditions, i.e., when the moisture transport is above its median value (see Methods and Fig. 2a). In 2011 to 2022, the absolute mass of ice melted in moist conditions increased substantially compared to 1981 to 2010 (9 ± 7 Gt yr$^{-1}$), while the melt in dry conditions, i.e., on all other melt-season days, stayed stable (see Fig. 2a). Moist conditions occurred more often ( +12 ± 9 d per melt season) in 2011 to 2022, compared to 1981 to 2010. Moist conditions are closely related to ARs: for 81% (confidence interval: 78 to 83%) of the moist conditions we find an AR in the proximity of Novaya Zemlya (see Methods; identified ARs from Mattingly et al.[18]). Especially in the high-mass-loss years, 2013, 2016, 2020, and 2022, 80% of the ice was lost during moist conditions. We find that, also, in 2018, which did not qualify as high-mass-loss year because of strong snowfall (16 ± 2 Gt), there was 27 Gt melt during moist conditions, roughly equal to 2013. In 2011 to 2022, the moisture transport was 12 ± 8% above its 1981 to 2010 average with its peak values in 2016 and 2018. The yearly snowfall increased insignificantly (1.8 ± 2.6 Gt yr$^{-1}$, $p$ = 0.14) in 2011 to 2022 compared to 1981 to 2010.

We note preceding changes of sea-ice concentration (SIC) and sea surface temperature (SST), visible in Fig. 2d. A low SIC and a high SST could facilitate that more energy arrives at the ice cap. Especially for 2016 and 2020, the SIC and SST were exceptionally low and high, respectively. We show a comparison of the skin temperature and the 2 m-air-temperature in Supplementary Fig. S8. However, it is not in the scope of the current study to disentangle how much the SIC and SST respond to the moisture import, how much they precondition its arrival at the ice cap, and how much they directly contribute to the heat flux anomalies.

### Foehn events

AR-induced foehn events are known to strongly promote surface melt in Greenland[16,18,24] and the Antarctic Peninsula[25]. The energy budget impact depends on the topography. In the case of northeast Greenland, the lee downslope atmospheric subsidence promotes cloud-clearance, thus enhancing downward shortwave radiation, amplifying surface melt. For the Antarctic Peninsula, the cloud-clearance effect has also been reported as surface heating process, except for situations where clouds can pass the mountain range and continue blocking direct sunlight. Evaluating the MAR downward shortwave radiation (see Fig. 3), we find that persistent clouds is the prototypical case for Novaya Zemlya. The distinction may be that Novaya Zemlya is too small to have a continental cloud effect like Greenland or Antarctica. According to CARRA (see Methods), the clouds over Novaya Zemlya are mainly situated between 850 and 700 hPa and can pass the mountain range, which has a 100 km-long section of elevations between 800 and 1000 m and is otherwise, with a few exceptions of limited extent, lower than that.

We categorize the melt depending on the wind direction (see Methods) and the magnitude of the moisture transport, to analyze connections between melt and moisture transport. We consider the spatial distribution of melt anomalies and anomalies of SEB components, shown in Fig. 3. The energy anomalies in the MAR output indicate a surplus of surface-directed sensible heat transfer on the leeward side in both moist and dry conditions. In moist conditions, a surplus of downward longwave radiation on the windward side is compensated by a deficit of the downward shortwave radiation. In dry conditions, the roles of short- and longwave radiation change: there is a surplus of downward shortwave radiation on the lee slopes—indicating cloud-clearance—that is compensated by a downward longwave radiation deficit. The surface latent heat flux anomalies are negative on the lee slopes, except for the case of moist easterlies. We associate the

negative anomalies with surface-melt-induced evaporative cooling. The positive latent heat anomalies on the lee-slope in moist easterly conditions are nevertheless too small an energy source to explain the observed mass loss.

The observed melt and surface energy anomaly patterns are more pronounced for easterly winds than for westerlies. We identify two factors that play a role: first, at 850 hPa the easterlies are 2 K warmer on average than westerlies in dry and in moist conditions. Second, we find that because of the orientation of the Novaya Zemlya terrain, and because moist conditions are less common for winds from the north, the average directions of westward and eastward winds lead to distinctly different incidence angles. The orientation of various extents of the slightly arc-shaped mountain range as azimuth bearing is between 30 to 70°. We pay special attention to the roughly 350 km long part, continuously higher than 500 m, with a orientation of 50° from (74.9°N, 57.8°E) to (76.6°N, 67.5°E). The moisture-transport-weighted average moist easterly winds arrive from 152°—impinging the terrain perpendicularly (10° incidence) on the inner side of the arc-shape. The westerly counterparts arrive from 291° and are inclined by roughly 30° relative to the terrain, coming from the outer arc side. The wind component that pushes the air over the mountain range is roughly 10% smaller than for the average easterlies. The incidence angles of dry westerly and easterly winds are roughly 20° and 0°, respectively. There is no significant difference ($p$ = 0.26) in the average moisture transport for westerly and easterly winds in moist conditions (149 ± 3 kg m$^{-1}$ s$^{-1}$ and 152 ± 5 kg m$^{-1}$ s$^{-1}$, respectively). In dry conditions, westerlies transport 60 ± 1 kg m$^{-1}$ s$^{-1}$ moisture and easterlies 51 ± 2 kg m$^{-1}$ s$^{-1}$. Lacking a tested, automatic foehn classification algorithm, quantifying the probabilities that certain conditions will trigger foehn winds cannot be included in this study. However, we hypothesize based on the current results that the pressure pushing air masses across the mountain range is respectively reduced (enhanced) because the open side of the arc-shaped mountain range points away from westerly (toward easterly) winds, leading to cloud development under easterly winds with stronger condensation-driven downward longwave radiation (Fig. 3e), reduced downward shortwave radiation (Fig. 3i), and more warming on the lee-slope (Fig. 3c).

In the CARRA data, we find that 83% (45 of the 54) days on which MAR calculates more than 1 Gt melt were accompanied by foehn winds (see Methods and Supplementary Table S3). Exemplary, in Fig. 4, we show the foehn event from 30 July to 3 August, 2020, a period associated with easterly winds. Panel (a) illustrates the higher air temperatures on the lee slope, panel (b) the lee slope downdrafts, and panel (c) and (d) the humidity gradients. The median 950 hPa-temperature difference between the leeward Barents and windward Kara Sea side evaluated in grid cells with average surface elevations between 380 and 480 m, roughly corresponding to the 950 hPa-pressure level, is 4 K. The moist-adiabatic isentropes indicate stably stratified lower atmosphere layers, especially on the leeward side. The same stratification is evident for cases with westerly winds, confirming that the dominant heat source is not the sea surface. The upward air motion following the downdraft on the lee-slope in panel (b) is a hydrolic jump, with the kinetic energy being converted to potential energy[26]. Similar to Fig. 4, we show averages of the 45 high-melt days with foehn winds for westerlies and easterlies in the Supplementary Figs. S9 and S10, respectively.

### Discussion

Here we have shown that in the years 2013, 2016, 2020, and 2022, particularly strong surface melt majorly contributed to the mass loss of Novaya Zemlya's glaciers. During atmospheric rivers episodes, surface melt is focused on the leeward side of the island, driven by foehn winds that deliver high fluxes (50 W m$^{-2}$) of additional sensible heat rather than, like in other regions, enhancing the downward shortwave radiation by cloud-clearance. The results highlight importance of

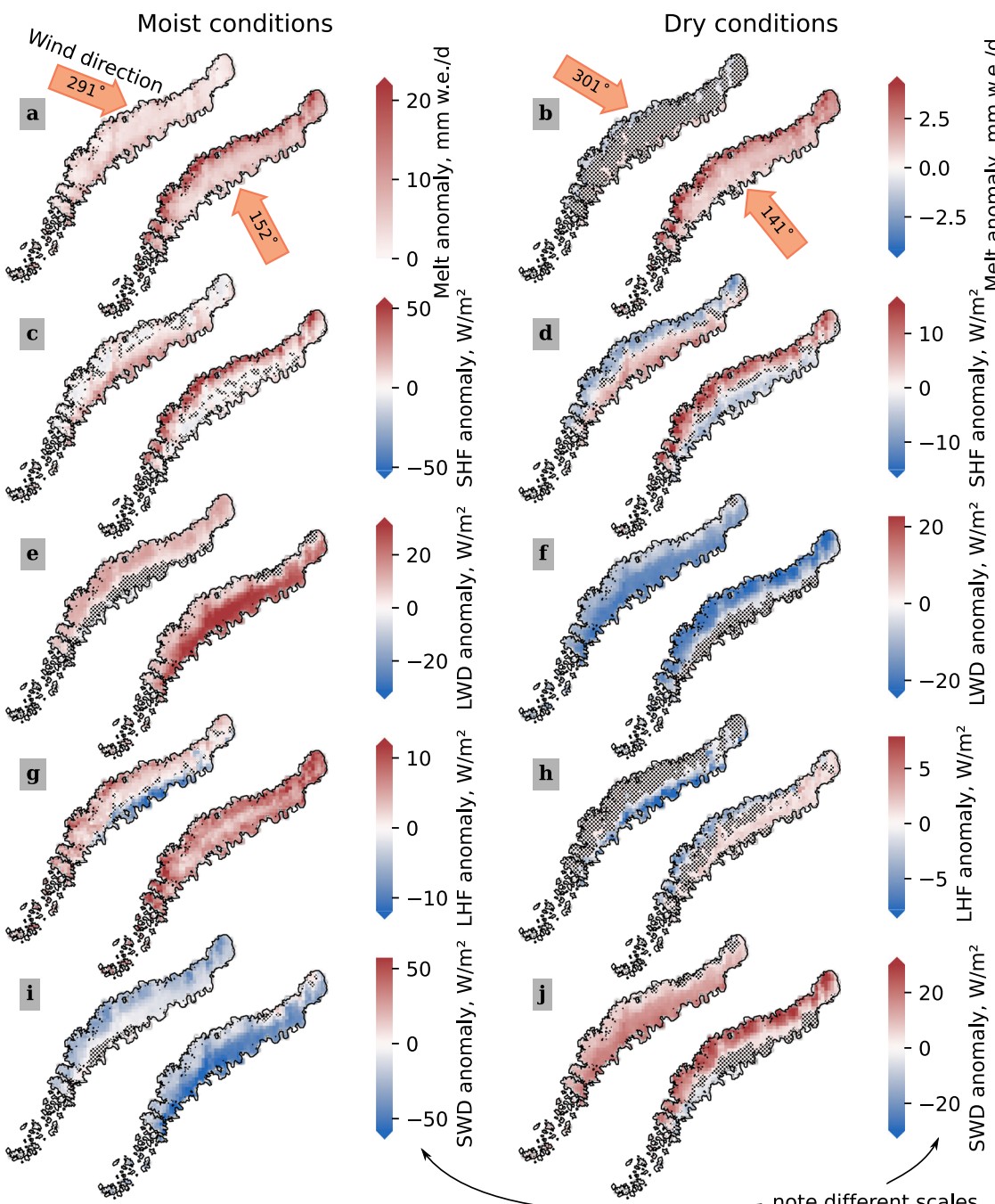

**Fig. 3 | Surface melt and energy flux anomaly maps. a, b** Spatial distributions of daily melt water production, (**c, d**) sensible heat flux (SHF), (**e, f**) downward long-wave radiation (LWD), (**g, h**) latent heat flux (LHF), and (**i, j**) downward shortwave radiation (SWD) anomalies with regard to 1981 to 2010, stippled where insignificant ($p \geq 0.05$). The vertical columns distinguish moist or dry conditions (see Results: moisture import into the region in recent times: over the 2011 to 2022 period, the correlation between the average moisture transport and the yearly melt was $r = 0.63$, while it was uncorrelated ($r = 0.07$) in 1981 to 2010.

Importance & evolution of moisture imports). Of each pair, the left image illustrates the average over days with westerly winds, the right one averages days with easterly winds. The orange arrows in the first row, (**a, b**), indicate the moisture-transport-weighted wind direction azimuth bearing at 850 hPa.

Additionally, the increased melt darkens the glacier surface, leading to a surplus of absorbed downward radiation (the melt-albedo feedback[27]). The melt-season broadband albedo decreased by 0.02 on average in 2011 to 2022 compared to 1981 to 2010 (see Supplementary Fig. S12). If compared to a hypothetical scenario with a constant albedo of the climatological average, the melt-albedo feedback contributed

roughly an additional 10 W m⁻² from shortwave radiation on average during the melt seasons of 2011 to 2022 compared to 1981 to 2010 and a maximum of additional 25 W m⁻² in 2020 (see Fig. 2b and Methods).

For the interpretation of the larger mass balance deficit along the Barents Sea compared to the Kara Sea, the results suggest that foehn events are the key-factor. While previous studies[12,13,15] pointed to the rapidly warming (0.08 K yr⁻¹ [28]) Barents Sea, the present results show that westerly winds, passing the Barents Sea before Novaya Zemlya, do not lead to excessive melt on the Barents Sea side, nor is the ice lost predominantly by tidewater glaciers. It is the easterlies that drive the

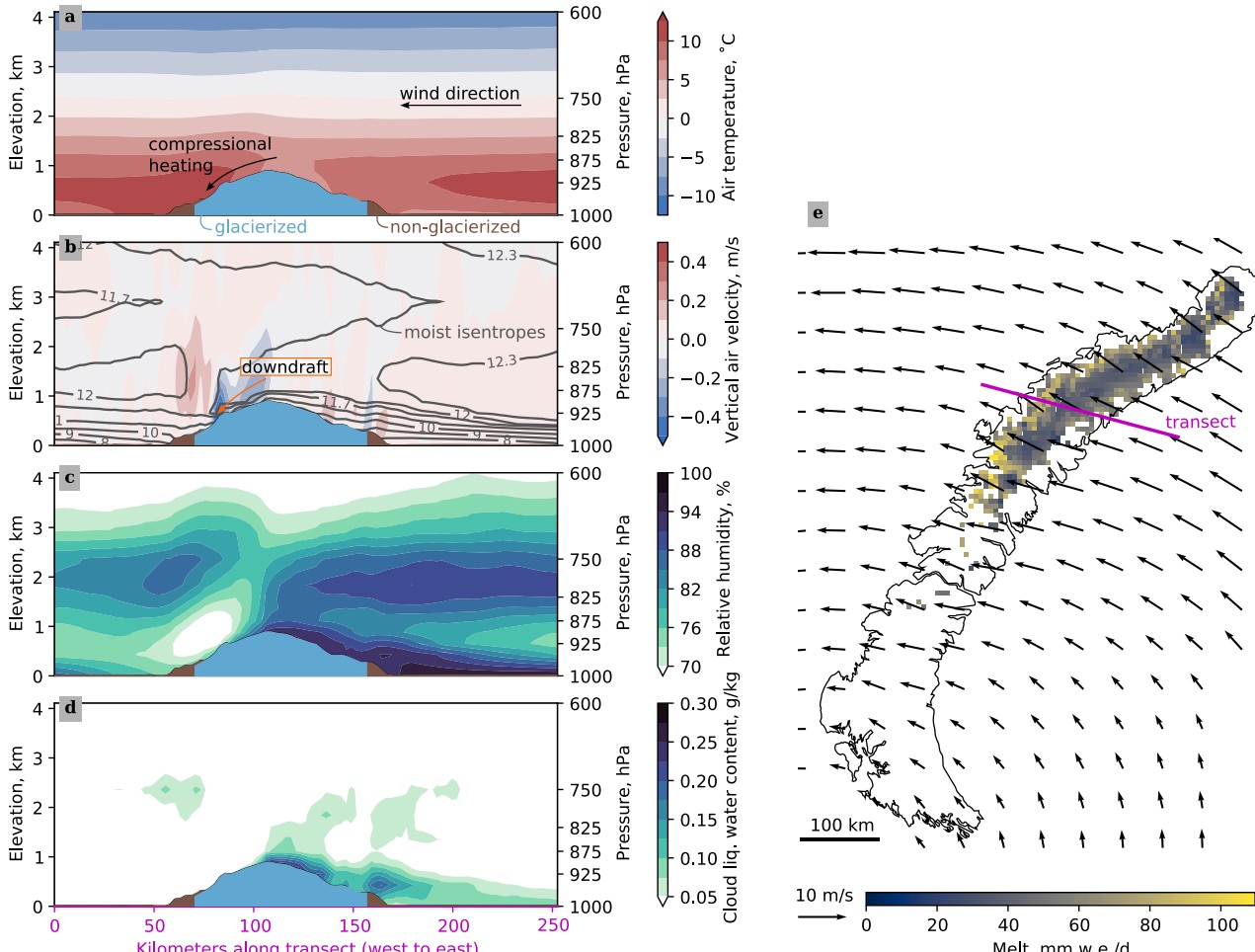

**Fig. 4 | Atmospheric cross section during foehn event.** The panels show average Copernicus Arctic Regional ReAnalysis (CARRA) (**a**) air temperature, (**b**) vertical air velocity, (**c**) relative humidity, and (**d**) cloud liquid water content from 6 am UTC on 30 July until 0 am on 4 August, 2020, along the transect highlighted in panel (**e**) from west to east. In panels (**a**–**d**), the brown and blue colored shapes indicate land and glacierized areas, respectively. The contour lines in panel (**b**) illustrate isotherms of the potential moist-adiabatic temperature in ·C. **e** illustrates the average CARRA wind speeds over the same period at 850 hPa, the average regional atmospheric model MAR daily melt water production from 30 July to 3 August, and the coastline of Novaya Zemlya.

melt on the Barents Sea side, in moist and in dry conditions. Theoretical snowfall surpluses on the Kara side, that would increase the albedo and reduce melt from absorbed shortwave radiation, are not expected because the increase of moisture transport would result in rainfall[29]. We, indeed, find that the albedo anomalies are more negative along the Kara coast (see Supplementary Fig. S12) because the moisture is precipitated as rain. Further, while the long-term average difference of area-specific mass changes in CryoSat−2 observations between land- and marine-terminating glaciers are in line with the discharge estimates of Kochtitzky et al.[11] and earlier studies[14,15,30], for the high-mass-loss years we found no termination type dependence of yearly area-specific change rates (see Supplementary Fig. S4). While there could be a potential dynamic response to surface processes[13] for the years with average losses, the area-specific comparison shows that such coupling was not a major contributor to the recent large mass losses.

Glaciers on Novaya Zemlya are projected to loose 27 ± 9% of their current volume until 2100 under RCP4.5[3]. However, the projected loss is based on input data of general circulation models (GCM). GCMs often fail to accurately simulate foehn winds, due to their coarse resolution[31]. Establishing robust relations between well-modeled observables and the melt from foehn winds could increase the precision of future projections. Only with accurate

projections of the future AR frequency and intensity could they offer a reliable predictor for present and future surface melt on Novaya Zemlya. An attractive, but costly, solution could be to embed regional climate models for glacierized regions in GCMs for better taking into account the role of local atmospheric circulations for surface melt, expressed before to better account for effects mitigating the melt-elevation feedback[32].

## Methods

### Significance of differences

We decide whether or not results are significantly different from a given reference based on the result of a two-sided Welch's $t$-test. We use the significance threshold $\alpha = 0.05$, i.e., for probabilities lower (equal or greater) than 0.05, we conclude that the difference is significant (insignificant).

### ERA5

ERA5 is the global atmosphere reanalysis model in the fifth generation from the European Center for Medium-Range Weather Forecasts (ECMWF)[33]. It estimates the state of the atmosphere at hourly 31 by 31 km postings and 137 vertical levels by assimilating observations and has been extensively validated[34]. ERA5's core component is the elaborate Integrated Forecasting System (IFS) Cy41r2.

## MAR

We use version 3.14 of the regional climate model MAR which is 6-hourly forced by the ERA5 reanalysis at its lateral boundaries and run here at a spatial resolution of 6 km. Sea ice cover and sea surface temperature are also prescribed from ERA5 over the MAR ocean pixels. The MAR model consists of an atmospheric module[35] fully coupled to the Surface Vegetation Atmosphere Transfer (SVAT) scheme Soil Ice Snow Vegetation Atmosphere Transfer (SISVAT)[36]. The snow model included in SISVAT is based on CROCUS[37], resolving most of the processes impacting the snow temperature, density, liquid water content, and grain size; except the snow drift which is negligible and not switched on, here[38]. We use here the extensively validated set-up of MAR, calibrated over the Greenland ice sheet[39] and recently applied over the Arctic ice caps, including Novaya Zemlya[10]. The main improvements of MARv3.14 with respect to MARv3.11 used in Amory et al.[40] and Maure et al.[10] are:

- a full rewriting of MAR code in Fortran 90 which has allowed to correct some bugs in its clouds scheme
- conservation of water mass into the soil and snowpack at each time step impacting mainly the water fluxes simulated by MAR over tundra
- a continuous snowfall-rainfall limit for near-surface temperature between −1 °C (full snow) and 1 °C (full rain)
- the use of the radiative scheme from ERA5[41] with respect to the old one from ERA40[42]
- a maximum liquid water content into the snowpack of 7% at the top of the snowpack and 2% below the first meter of snow with a continuous transition between both those values

All of these improvements have been successfully validated over the Greenland ice sheet where they show to better compare with SMB measurements[39], satellite derived melt extent[43], and in situ atmospheric measurements[44].

## Input-output model

We calculate the Input-Output (I/O) mass budget by complementing the MAR SMB time series (input) with the decadal discharge estimate (output) from Kochtitzky et al.[11]. To acknowledge a transition between the estimates for the two decades, we interpolate the discharge $D(t)$ between $t_0 = 1$ January, 2000, and $t_1 = 31$ December, 2020, and assume constant discharge before and afterward.

$$D(t) = \begin{cases} 1.4 \text{ Gt yr}^{-1} & \forall\, t <= t_0 \\ 1.4 \text{ Gt yr}^{-1} + (t - t_0) \cdot 0.11 \text{ Gt yr}^{-2} & t \in [t_0, t_1] \\ 3.8 \text{ Gt yr}^{-1} & \forall\, t >= t_1 \end{cases} \quad (1)$$

We acknowledge that the simplicity of the chosen discharge model neglects possible inter-glacier and temporal variability, such as seasonal summer glacier speed up[45]. This is justified by the good agreement between the model and observational data from gravitational and geodetic measurements.

We estimate the $2\sigma$-uncertainty of the cummulative SMB to be the bigger of either 10% of the snowfall plus the rainfall, or 10% of the runoff. The discharge uncertainty $u_D$ is propagated from the uncertainties given in Kochtitzky et al.[11] under the assumption that the uncertainties of both decades are not correlated. The I/O mass balance uncertainty $u_{I/O}$, then, is:

$$u_{I/O} = \sqrt{u_{SMB}^2 + u_D^2} \quad (2)$$

$$u_{I/O}^{1981-2010} = \sqrt{4.0 + 1.5} \text{ Gt yr}^{-1} = 2.4 \text{ Gt yr}^{-1} \quad (3)$$

$$u_{I/O}^{2011-2022} = \sqrt{3.2 + 2.2} \text{ Gt yr}^{-1} = 2.3 \text{ Gt yr}^{-1} \quad (4)$$

## Glacier outlines, fronts, and ice divide

We use the glacier outlines and termination types from the Randolf Glacier Invertory (RGI) version 6.0. For the moisture import analysis and the comparison of modeled and observed albedo data in Results (Importance & evolution of moisture imports), as well as the average wind direction calculation in Methods (Wind direction), we consider only the largest basins that collectively constitute 80% of the glacierized area because of the model and reanalysis data's spatial resolution. For all other analysis, we consider all basins.

To compare the 2010 to 2022 mass change rates to Moholdt et al.[12] in Results (Mass loss over the past decades) we adopt their definition of fronts and of the ice divide. Glacier fronts are areas below 500 m. The ice divide is the separator between RGI basins along the Barents and Kara coasts buffered by 5 km above an elevation of 500 m.

## CryoSat−2

CryoSat−2 is a spaceborne nadir-looking Ku-band radar altimeter from the European Space Agency (ESA)[46]. When it passes over glaciated areas with rough topography, as is the case in Novaya Zemlya, it uses two antennas that provide interferometric capabilities and enable retrieving multiple elevation estimates around its ground track. We use this feature to derive geodetic glacier mass changes from swath-processing of ESA's freely available SARIn L1b data product at baselines D (until 21 August, 2021) and E (afterward). We retrieve surface elevation estimates by exploiting the phase difference between the received signal at both antennas to calculate the angle of incidence and use the coherence of both signals to detect flawed signals; we reconstruct up to 1024 points from which the signal was reflected at approximately every 310 m along CryoSat−2's ground track within a roughly 15 km-wide swath[47].

We collect the point data into 500 by 500 m grid cells, where the elevation of the static digital elevation model ArcticDEMv3.0[48] is larger than 10 m to avoid contamination from ocean signals. We derive the elevation change rate for each cell by a linear regression of the differences between ArcticDEM and the CryoSat−2 elevation estimates. When residuals lie outside three times the scaled median absolute deviations (MAD), they are recursively rejected as outliers. The scaling factor for the MAD is 1.48, which aligns this outlier-robust measure with standard deviations of a normal distribution[49]. We require the remaining data to span more than 5 yr. Further, we reject trends below −15 m yr$^{-1}$, larger than 5 m yr$^{-1}$, and those of which the 95%-confidence interval exceeds 0.5 m yr$^{-1}$. After we fitted the trends per grid cell, we calculated the median residuals of 3-monthly rolling windows to obtain the monthly variations.

Voids are filled by hypsometric interpolation using a weighted-least-square fitted third-order polynomial[50] for each RGI-basin separately. Glacier basins that are smaller than 10 km$^2$ or contain less than 20 grid cells to which a trend could be fitted, are treated collectively in two groups, one for the Barents and one for the Kara coast. All of these small glaciers are land terminating. To acknowledge the large meridional range of our study area, we include linear horizontal dependencies in the fitted interpolation function for the collectively-treated glaciers[6]. Residuals outside three times the scaled MAD are treated as voids (non-iteratively).

We convert the volume trends and the intra-annual mass changes by a dual density model. Because Novaya Zemlya experienced sustained mass loss for a long period, we assume that a surface elevation decrease below the minimum value observed in the preceding record for each cell results from loss of glacial ice and that the associated lost volume has the density of $\rho_{ice} = 917$ kg m$^{-3}$. This assumption still permits the existence of a firn layer, which has a constant thickness for each grid cell. Above the observed minimum, we assume that surface elevation changes are mainly due to the gain or loss of young firn with a density of $\rho_{firn} = 650$ kg m$^{-3}$.

Because Ku-band radar penetrates dry snow[51], we derive higher-level products, i.e., the 2010 to 2022 mass balance and the yearly mass losses, based on the average surface elevations at the end of each melt season on 1 October to minimize the bias from snow-pack penetration. At the end of the melt season, the surface properties are most consistent—close to the glacier-air interface is a dense layer from refrozen melt water that strongly reflects the radar signal. We acknowledge that there can still be differences from year to year due to, e.g., substantial snowfall before 1 October, or an incomplete dense firn layer. Further, yearly evaluation reduces the propagation of systematic errors in CryoSat-2 data to a minimum[47].

We estimate the mass change uncertainty, in essence, following the choices earlier made by Jakob and Gourmelen[6]. We adopt the 2 km decorrelation length for elevation estimate uncertainties $u_h$, which results in roughly 5000 independent elevation estimates per time step. Furthermore, we assume that the uncertainty of the glacierized area $u_A/A$ is 5%, and that the uncertainty in the ice density $u_\rho^{\text{long-term}}$ is 60 kg m$^{-3}$ over the 2010 to 2022 period and $u_\rho^{\text{short-term}} = 150$ kg m$^{-3}$ for yearly changes, depending on the context[6,52]. We assume that long-term changes of the average penetration depth of the radar signal into the firn are sufficiently acknowledged by the density uncertainty. The resulting total uncertainty for the October mass anomaly estimates that are used to derive the long-term mass balance and the yearly changes is on average 6.8 Gt. For the long-term mass balance uncertainty $u_{\dot{M}}$ these assumptions yield:

$$u_{\dot{M}} = \sqrt{(A\rho u_h)^2 + \left(\dot{h}A u_\rho^{\text{long-term}}\right)^2 + (\dot{h}\rho u_A)^2} \quad (5)$$

$$= \sqrt{1.6 + 0.7 + 0.4} \text{ Gt yr}^{-1} = 1.6 \text{ Gt yr}^{-1} \quad (6)$$

### Short-term melt variability
As a measure for the short-term melt variability, we calculate the share of melt above a low-pass filtered version. In detail, we apply a moving Gaussian 61-day window with a 31-day standard deviation to the daily melt amounts. We, then, divide the melt above the low-pass filtered version by the total (see visualization in Supplementary Fig. S6).

### Climatological average and anomalies
We calculate anomalies by comparing a given value to the reference period, 1981 to 2010. In detail, we first calculate the median value over the reference period per day-of-year. We, then, calculate the moving mean (centered 15-day window) to obtain a low-pass filtered version of the day-of-year medians. For spatially resolved data, we proceed as described for each grid cell.

### Atmospheric rivers and moisture transport
We use the AR identification and moisture transport values of the dataset described in Mattingly et al.[18]. The earlier study calculated the moisture transport from the Modern-Era Retrospective analysis for Research and Applications, Version 2 (MERRA-2) data (resolution: 0.5 by 0.625°) and identifies ARs based on the moisture transport. North of 66.56°, patches with a moisture transport > 150 kg m$^{-1}$ s$^{-1}$ and above the 85th climatological percentile classify as ARs if their length is >1500 km and their length-to-width ratio is >1.5. Using a different detection algorithm will lead to differences in the detection, the extent, and the duration of ARs[53]. O'Brien et al.[53] calls for additionally using the vertical dimension for the identification of atmospheric rivers. This, however, has not been achieved, yet.

In the present study, we find different melt-energy compositions for events with larger and smaller moisture transport, but regardless of the elongation of the large-moisture patches. We choose to group our data into the class of moist and dry conditions where we use the melt season median as threshold. We downsample the 3-hourly data to daily averages. Spatially we aggregate the moisture transport data over all grid cells that touch any of the 80% largest (see Methods: Glacier outlines, fronts, and ice divide) RGI-basins. We avoid smaller basins because of the coarse 0.5 by 0.625° resolution.

To study the relation between moist conditions and ARs passing close-by, we calculate the correlation coefficient $r$ of the number of moist days and the number of days with ARs passing in a 50 km-radius during the melt season of each year (1980 to 2022). We find $r = 0.81$ (stated in Results); Supplementary Fig. S11 shows the number of classified days per year for both methods. We further calculate the conditional probability of the presence of ARs in the case of moist conditions. To mitigate the impact of temporal correlation, we evaluate the probability for each week and average afterward. We derive the 95%-confidence interval from bootstrapping with 10 000 iterations.

### Wind direction
We calculate wind directions from daily averages of 3-hourly ERA5 wind speeds at 850 hPa over the 80% largest (see Methods: Glacier outlines, fronts, and ice divide) RGI-basins. We define westerlies and easterlies as the two solid angles 245°–35° and 65°–215°, respectively. With this definition, westerlies come from the Barents Sea and easterlies from the Kara Sea.

### CARRA
The Copernicus Arctic Regional ReAnalysis (CARRA) is a reanalysis product of the Copernicus Climate Change Service (C3S)[54,55]. Its core component is the weather forecast model HARMONIE-AROME[56]. CARRA provides 3-hourly data at 65 vertical levels and a 2.5 by 2.5 km horizontal resolution.

**Foehn wind classification.** We use CARRA data, among other ends, to identify foehn winds on days where the MAR surface melt exceeds 1 Gt. We inspect the daily averages of the air temperature, the moist-adiabatic isentropes, the vertical air velocity, the relative humidity, and the cloud liquid water content for vertical profiles along a transect. The transect is the same for all concerned days and was chosen such that it crosses the main mountain ridge from west to east roughly in the average direction of westerly and easterly winds. We show the profiles exemplary for the first day that exceeds the melt threshold in Supplementary Fig. S14.

We classify days into those characterized by foehn winds and those that are not, based on five criteria. With foehn winds, we expect (1) the air temperature to be larger on the leeward than on the windward slope, (2) the moist-adiabatic isentropes to be rather horizontal on the windward side but descend on the leeward slope, (3) the air to descend on the leeward slope, and (4) the relative humidity and (5) the liquid cloud water content to drop from the windward to the leeward slope. If three out of the five criteria are fullfilled, the concerned day is rated to be characterized by foehn winds.

### Albedo feedback contribution
We calculate the MAR modeled share of downward shortwave radiation SWD that was additionally absorbed by the surface compared to the climatological average as the albedo anomaly $\Delta\alpha$ times SWD.

$$\text{netSW} = \text{SWD} \cdot (1 - \alpha) \quad (7)$$

$$= \text{SWD} \cdot (1 - \alpha_{\text{clim}}) - \text{SWD} \cdot \Delta\alpha \quad (8)$$

$$= \text{netSW}_{\text{clim}} + \Delta \text{netSW} \quad (9)$$

$$\Rightarrow \Delta \text{netSW} = -\text{SWD} \cdot \Delta\alpha \quad (10)$$

For 2000 to 2022, we compare melt-season averages of the MAR modeled albedo $\alpha_{MAR}$ to the average of the broadband white- and black-sky albedo $\alpha_{MODIS}$ from the MCD43A3.061 MODIS dataset[57] freely provided by NASA's Land Processes Distributed Active Archive Center (LP DAAC). We find a correlation coefficient of $r = 0.91$ ($p \approx 1 \times 10^{-9}$) between the modeled and observed albedo data and the following relation from least-squares regression (see Supplementary Fig. S13):

$$\alpha_{MAR} = (0.72 \pm 0.07) \cdot \alpha_{MODIS} + (0.20 \pm 0.05) \tag{11}$$

## Data availability

Glacier mass change time series, time series of modeled surface mass fluxes, moisture transport and AR time series, and CryoSat−2 derived surface elevation trend maps generated in this study have been deposited in the 4TU.ResearchData database and are available under https://doi.org/10.4121/10753234-8bf5-4f8a-b427-2eec0b3af060. Additionally, large intermediate products, including CryoSat−2 derived products, MAR model output, and GRACE/GRACE-FO derived products will be temporarily made available upon request to the corresponding authors. ERA5 and CARRA data are available at https://cds.climate.copernicus.eu. The glacier outlines are available at https://nsidc.org/data/nsidc-0770/versions/6. The MODIS data are available at https://lpdaac.usgs.gov/products/mcd43a3v061. Source data to reproduce Figs. 1 and 2 are provided with this paper. Source data are provided with this paper.

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

## Acknowledgements

J.H. and B.W. were supported by NWO VIDI grant `016.Vidi.171.063`. The authors would like to thank Kyle Mattingly for providing the AR classification data, Sophie de Roda Husman for providing the MODIS processing scripts, David Rounce for providing regional projection results, ESA for providing CryoSat–2 L1b data, the ECMWF for providing ERA5 data, and the Polar Geospatial Center for providing ArcticDEM under NSF-OPP awards `1043681`, `1559691`, and `1542736`. CARRA data originate from the EU C3S contract `2017 C3S_322_Lot2_METNO SC2`.

## Author contributions

Conceptualization: J.H. and B.W.; data processing: J.H., B.W., and X.F.; visualization: J.H., B.W., and J.E.B.; investigation: J.H. and B.W.; inter-pretation: J.H., B.W., X.F., J.E.B., and I.A.G.; writing: J.H., B.W., X.F., J.E.B., and I.A.G.

## Competing interests

The authors declare no competing interests.
