## [Peer Review File · Nature Communications]

Atmospheric-river-induced foehn events drain glaciers on
Novaya ZemlyaREVIEWER COMMENTS

Reviewer #1 (Remarks to the Author):

Review: Atmospheric-river-induced foehn events drain glaciers on Novaya Zemlya

The manuscript explores the mechanisms behind recent glacial melt on Novaya Zemlya with a combination of climate modeling, space-based measurements, and reanalysis datasets. The authors find that a large fraction of melt occurs during moist atmospheric conditions, frequently accompanied by an atmospheric river. The mechanism of melt is primarily the resultant foehn winds that occur on the leeward side of the glacier, and unlike foehn winds in some other locations, the largest contributor to melt is the transfer of sensible heat from the atmosphere to the ice.

The manuscript is well-written and easy to follow. The figures are clear and are mostly sufficient to back up the hypotheses put forth by the authors. The topic of atmospheric rivers, foehn winds, and glacial melt has received considerable attention in recent years, and the focus on the Russian high Arctic is a welcome contribution to the emerging topic. My comments to improve the manuscript are mostly minor and outlined below:

1. Most importantly, can you quantify and show why some ARs drive Foehn events, but others do not? What percentage of ARs result in Foehn events? Are Foehn events more or less likely with ARs from a particular direction? Of a particular moisture content?
2. Can you provide a version of Figure 3 that shows only those grid cells with statistically significant changes? A supplemental figure should suffice.
3. Though the case study example in Figure 4 is effective for demonstrating the processes that drive melt during the Foehn event, can you provide a composite plot of Foehn events as well? One set of figure panels for easterly events and another for westerly events. This will showcase the robustness of the melt processes during these Foehn events. A supplemental figure should suffice.
4. Can you also provide a map of the melt on AR days? My understanding is that this isn't quite what is being shown in Figure 3 because not all high moisture days are AR days. A supplemental figure should suffice.
5. How were the Foehn events identified in CARRA?
6. Please include a discussion regarding uncertainty in AR detection based on the choice of algorithm. See, for example, O'Brien et al. (2020): <https://doi.org/10.1175/BAMS-D-19-0348.1>
7. I was a bit confused by your text in the Introduction that states "ARs reach the Barents and Kara Seas with increasing frequency in the months Nov, Dec, and January...". This isn't the season you focus on in the manuscript. Have there been any other studies characterizing AR events during the melt season of June-Sept?

Reviewer #2 (Remarks to the Author):

In this paper, the authors have done an impressive job combining a wide range of techniques and datasets, to show that the increased mass losses from glaciers on Novaya Zemlya are due to an increase in Atmospheric Rivers and related conditions, rather than increased discharge from marine-terminating glaciers or ocean-induced temperature changes, as has previously been argued/shown.

I really enjoyed reading this paper, despite the bulk of it being outside of my normal area of expertise. Overall, I found the methods to be sound and well-explained, and the results displayed in a way that helps demonstrate the main argument of the study. As such, my comments are largely minor, with the exception of one more general comment shown below. I think that the paper fits within the scope of Nature Communications, and I see no issues with its being accepted, with the caveat that I am not a climate modeller by training or background.

General

What is the frequency of moist conditions during the melt season vs during winter season? You have shown, in Fig. S7, that there appears to be an increase in the number of days of moist conditions during the melt season. Fig. 2c additionally shows that there is no appreciable difference in the snowfall component of SMB over the period 1981-2022, which seems to rule out an increase in snowfall helping to offset some of the increased loss due to melt, but it doesn't necessarily address the potential for increased liquid precipitation (and therefore melt) outside of the "melt season". With an increase in the number of wintertime rain events in the Arctic (e.g., Bintanja and Andry, 2017), one might expect that an increase in moist conditions outside of the melt season could be at least partly responsible for the observed increase in mass loss.

I recognize that a thorough investigation of this is likely at least a whole study in its own right, but I think it should be possible to use your observations and data to show whether there are any appreciable changes in moist conditions during the winter season. This might also help further solidify the point you make on page 9 at line 24 ("We could not find evidence [...]") - at least, it might help confirm that the bulk of the observed increase in melt is occurring due to the changes observed during the melt season.

Specific comments

p. 1, l. 19-21: Here, I think you can better motivate this study by (a) switching the order of the last two sentences ("The increased ice loss [...] of many [5]. Despite this, confidence [...]"), followed by a brief summary of some of the reasons why these projections are incomplete, and how what you have shown here points towards one area to focus on: namely, that most (all?) projections, especially at the global scale, do not adequately account for the impacts of extreme weather events.

p. 1, l. 25: is this based on the RGI v6, or on the reference to Millan et al. (which used RGI v6)?

Figure 1: What are the colors of the annual MB patches meant to indicate? This should be included in a legend or color bar along the side.

p. 3, l. 1-3: "difference modeled the climatic surface mass balance": do you mean to say that you "difference the modeled climatic mass balance [...] and published ice discharge estimates"?

p. 4, l. 7-8: I'm not quite sure I follow this - do you mean that in the year 2014, these fluxes (sensible heat, incoming long-wave radiation, latent heat) were all lower than the 1981-2010 average, which contributed to this being a positive MB year? Or was the positive MB year in 2014 in spite of these fluxes being higher than the 1981-2010 average?

p. 4, l. 17: could this be shown in the supplemental material?

p. 4, l. 23: is this difference comparing 2011-2022 with 1981-2010?

p. 4, l. 27: how much snowfall was there during this year (2018)?

Figure 2: is it possible to somehow indicate the number of AR episodes here?

Figure 3 caption: "The arrows [...] categories, moist and dry conditions, in shades of orange (deep and pale, respectively)". Given that the two columns of the figure already differentiate between moist and dry conditions, I'm not sure that you need to further indicate this using the arrow colors here - I think you can leave these as a single color and end the caption at "850 hPa".

p. 7, l. 13: this is in moist easterly conditions, right? As I interpret Fig. 3, the LHF anomaly is negative on the lee slope under moist westerly conditions, and positive on the lee slope in moist easterly conditions.

p. 9, l. 17-18: are you able to quantify the MB or melt impact of this additional SW radiation?

p. 10, l. 16: why is the snow drift not switched on? Is there an indication that drifting snow does not play a large role in the snowpack here?

p. 10, l. 25: can you briefly state the difference between these two radiative schemes?

p. 12, l. 14: do you have an indication of how much this might be overestimating the mass, by undercounting the firn? If I understand this correctly, you assume that anything below the previous minimum elevation value is ice - in effect, that the previous minimum value represents ice. This is reasonable for cells at lower elevations, but what about at higher elevations? If I have misunderstood this, and "minimum value" and "preceding record" are not somehow referring to surface elevation, I am happy to be corrected here.

p. 12, l. 32: how have you derived u_{ρ} here? In the preceding paragraph, you state your values for $u_{\rho}^{\text{short-term}}$ and $u_{\rho}^{\text{long-term}}$, but do not describe how you combine these to get u_{ρ} .

p. 14, l. 7: it might be clearer to write this as $\Delta \text{netSW} = -\text{SWD} \cdot \Delta \alpha$ - as written, it is easy to miss the \cdot , making it seem like this is $\text{SWD} - \Delta \alpha$.

p. 15, ref. 5: still "in press"?

p. S1, Table S2: these values are, presumably, averaged over different areas - can you include those areas here, to give an idea of the partitioning/impact on the total mass balance from each area (ice divide, lower elevations, etc)?

p. S4, Fig. S3: can you show the outline of the divide area on this map?

References

Bintanja R and Andry O 2017 Towards a rain-dominated Arctic Nat. Clim. Change 7 263–7.
<https://www.nature.com/articles/nclimate3240>

Reviewer #3 (Remarks to the Author):

see the attached documents

Haacker et al. report a possible mechanism controlling the recent rapid mass loss of glaciers in the Russian High Arctic by analyzing a wide range of datasets. They argued that the recently increased ice mass loss could be explained by foehn winds accompanied by extreme energy imports from atmospheric rivers, which were previously believed to be caused by dynamic calving glacier behaviors. I believe the study has great potential to be published in a high-impact journal and attract significant interest from researchers across a wide range of research fields.

Although there are several concerns that I believe the authors need to explain. Since there were no line numbers on the manuscript, my specific comments were written directly in the manuscript. I hope my comments help enhance the significance of the study.

Major comments

The study assumed nearly constant ice discharge during 2000-2010 and 2010-2020 as quantified by the previous study. But how is this assumption plausible? For example, Melkonian et al. (2016) reported that ice flow speeds vary substantially from year to year in the region, which could potentially explain the mass loss.

Somewhat related to my first point, I understand that the authors are explaining that the surface mass balance dominates the mass budget because of the similar annual specific mass balance between land- and marine-terminating glaciers. However, how reliable are the mass changes derived from altimeter for the glaciers which has a relatively narrow ice front? The authors explained that the mass change was quantified using a 500 x 500 m gridded dataset, while the glaciers have several kilometers width at the lower elevations. This limited resolution of the data may underestimate the mass change of the dynamic thinning of the marine-terminating glaciers. Maybe you could compare your altimetry-derived specific mass change with the recently published geodetic mass change in the region (for example by Sommer et al. 2022)?

3. Why and how do atmospheric rivers frequently occur in recent years in the region? Sea ice concentration and SST are indicated in Figure 2d but not compared with your results. They appear to be closely related to extreme melt years in 2016 and 2020. It is important to discuss how these factors are related to the occurrence of atmospheric rivers and extreme ice melting with relevant previous studies.

Specific comments

Please find my comments on the PDF files.

References:

- Melkonian, Andrew K., et al. "Recent changes in glacier velocities and thinning at Novaya Zemlya." *Remote Sensing of Environment* 174 (2016): 244-257.
- Sommer, Christian, et al. "Brief communication: Increased glacier mass loss in the Russian High Arctic (2010–2017)." *The Cryosphere* 16.1 (2022): 35-42.

Editorial Note: In their review of the first version of this manuscript, reviewer 3 added their comments to the manuscript file. These comments, excluding minor textual revisions, have been copied into this Peer Review File.

Comments in Main Text

Comment 1 Could it be more specific? For example, Melkonian et al., 2016 is rather showing a strong increase of ice speed in recent years?

Comment 2 It sounds a bit awkward choice of word.

Comment 3 Where this number come from?

Comment 4 I could not possible to follow how you calculate this number.

Comment 5 You could also compare your result with more recent geodetic mass change (Sommer et al., 2022). It is also important to compare the number for each basin. I concern the altimetry derived mass loss underestimate the mass loss for the calving glaciers (see my major comment).

Comment 6 how those years relate to sea ice and SST in the region as well as large-scale climate variabilities? It would be nice if the authors discuss it somewhere in the manuscript.

Comment 7 Looks like this panel was never cited in the manuscript, but it would be interesting to compare with your results.

Comment 8 I found this plots are hard to follow because lots of information packed in. You should think of labeling panels with a, b, etc and cite specific panel in the main text.

Comment 9 This paragraph was a bit hard to follow because no panel label in the Figure 3.

Comment 10 How the model performs over Novaya Zemuriya but not over Greenland? You should rather evaluate your model input and output with ground truth obtained at Novaya Zemlya.

Comment 11 How much this assumption plausible?

Comments in Supplement

Comment 1 You could indicate the 500 m elevation contour to highlight the ice front and ice divide regions?

Comment 2 Could you specify which marker represents which year by color or text especially for 2013, 2016, 2020 and 2022?

Comment 3 What is the shaded area means?

Responses to Reviewers' Comments for Manuscript NCOMMS-24-00742

Atmospheric-river-induced foehn events drain glaciers on Novaya Zemlya

Addressed Comments for Publication to

by

J. Haacker*, B. Wouters*, X. Fettweis, I.A. Glissenaar, J.E. Box

Credit: We use the "Review Response Template", created by Karl-Ludwig Besser.

Authors' Response to Reviewer 1

Comment 1

Results: Foehn events

Most importantly, can you quantify and show why some ARs drive Foehn events, but others do not? What percentage of ARs result in Foehn events? Are Foehn events more or less likely with ARs from a particular direction? Of a particular moisture content?

Response: Thank you for the comment.

The shape of the mountain range plays a role for the probability that foehn winds are triggered by westerly or easterly winds. We find that in similar conditions, the melt and energy balance patterns during easterly winds show more pronounced patterns that point to foehn winds. Unfortunately, we are not aware of a method to automatically evaluate the occurrence of foehn winds for all time steps. Our stack proof study of the high melt events is the best we can do without having to develop new methods, which would certainly exceed the scope of this brief communication. We, now, make this explicit by adding:

Lacking a tested, automatic foehn classification algorithm, quantifying the probabilities that certain conditions will trigger foehn winds cannot be included in this study. However, we hypothesize based on the current results that the pressure pushing air masses across the mountain range is respectively reduced (enhanced) because the open side of the arc-shaped mountain range points away from westerly (toward easterly) winds, leading to cloud development under easterly winds with stronger condensation-driven downward longwave radiation (Fig. 3.e), reduced downward shortwave radiation (Fig. 3.i), and more warming on the lee-slope (Fig. 3.c).

Comment 2

Figure 3

Can you provide a version of Figure 3 that shows only those grid cells with statistically significant changes? A supplemental figure should suffice.

Figure 1: (a)-(b) Spatial distributions of melt, (c)-(d) sensible heat flux (SHF), (e)-(f) downward longwave radiation (LWD), (g)-(h) latent heat flux (LHF), and (i)-(j) downward shortwave radiation (SWD) anomalies, stippled where insignificant ($p \geq 0.05$). The vertical columns distinguish moist or dry conditions (see Results: Importance & evolution of moisture imports). Of each pair, the left image illustrates the average over days with westerly winds, the right one averages days with easterly winds. The orange arrows in the first row, (a)-(b), indicate the moisture-transport-weighted wind direction azimuth bearing at 850 hPa.

Response: Thank you for the comment.

We agree that this adds important information and adapted Fig. 3 in the main article; here shown as Fig. 1 (stippled regions are insignificant ($p \geq 0.05$)).

Comment 3

Figure 4

Though the case study example in Figure 4 is effective for demonstrating the processes that drive melt during the Foehn event, can you provide a composite plot of Foehn events as well? One set of figure panels for easterly events and another for westerly events. This will showcase the robustness of the melt processes during these Foehn events. A supplemental figure should suffice.

Response: Thank you for the comment.

We agree that the suggested figures would be valuable and contribute to a more general understanding. We now show the produced figures in the supplement Figs. S9 and S10 and here in this rebuttal, in Figs. 2 and 3. We refer to the new figures by adding the following sentence at the end of Results: Foehn events.

Similar to Fig. 4, we show averages of the 45 high-melt days with foehn winds for westerlies and easterlies in the supplement Figs. S9 and S10, respectively.

Comment 4

How were the Foehn events identified in CARRA?

Response: Thank you for raising this question.

For those days, where MAR calculates more than 1 Gt surface melt (roughly 99th-percentile in melt-season, i.e., June to September; meaning that only the high melt days are analysed), we obtain and average the 3-hourly CARRA output. We, then, plot transects of the air temperature, the moist-adiabatic isentropes, the vertical air velocity,

Figure 2: In analogy to main article Fig. 4, this figure shows the average conditions at 12 UTC for days with westerly winds and MAR melt exceeding 1 Gt (see Table S3). The panels show the CARRA (a) air temperature, (b) vertical air velocity, (c) relative humidity, and (d) cloud liquid water content along the transect highlighted in panel (e) from West to East. In panels (a-d), the brown and blue colored shapes indicate land and glacierized areas, respectively. The contourlines in panel (b) show isotherms of the potential irreversible moist-adiabatic temperature in °C. Panel (e) shows the average CARRA wind speeds at 850 hPa, the average MAR melt, and the coastline of Novaya Zemlya.

Figure 3: Same as Fig. 2, but for easterly winds.

the relative humidity, and the liquid cloud water content. In the presence of foehn winds, we expect the air temperature to be larger on the leeward than on the windward slope, the moist-adiabatic isentropes to be horizontal on the windward side but descend on the leeward slope, the air to descend on the leeward slope, and the relative humidity and the liquid cloud water content to drop from the windward to the leeward slope. If any of those criteria are violated, we conclude that on average this day should not be considered a foehn event. Further, at least three of the five criteria have to be confirmed before we conclude that in average the day was characterized by a foehn event.

We now clarify this in the Methods section. We added the following:

Foehn wind classification

We use CARRA data, among other ends, to identify foehn winds on days where the MAR surface melt exceeds 1 Gt. We inspected the daily averages of the air temperature, the moist-adiabatic isentropes, the vertical air velocity, the relative humidity, and the cloud liquid water content for vertical profiles along a transect. The transect is the same for all concerned days and was chosen such that it crosses the main mountain ridge from West to East roughly in the average direction of westerly and easterly winds. We show the profiles exemplary for first day that exceeds the melt threshold in supplement Fig. S14.

We classify days into those characterized by foehn winds and those that are not, based on five criteria. With foehn winds, we expect the air temperature to be larger on the leeward than on the windward slope, the moist-adiabatic isentropes to be rather horizontal on the windward side but descend on the leeward slope, the air to descend on the leeward slope, and the relative humidity and the liquid cloud water content to drop from the windward to the leeward slope. If three out of the five criteria are fulfilled, the concerned day is rated to be characterized by foehn winds.

Comment 5

Please include a discussion regarding uncertainty in AR detection based on the choice of algorithm. See, for example, O'Brien et al. (2020): <https://doi.org/10.1175/BAMS-D-19-0348.1>

Response: Thank you for the suggestion.

We added the following in Methods: Atmospheric rivers and moisture transport.

Using a different detection algorithm will lead to differences in the detection, the extent, and the duration of ARs [1]. O'Brien, Payne, Shields, *et al.* [1] calls for additionally using the vertical dimension for the identification of atmospheric rivers. This, however, has not been achieved yet.

- [1] T. A. O'Brien, A. E. Payne, C. A. Shields, *et al.*, "Detection uncertainty matters for understanding atmospheric rivers," *Bulletin of the American Meteorological Society*, vol. 101, no. 6, E790–E796, Jun. 16, 2020. DOI: 10.1175/BAMS-D-19-0348.1.

Comment 6

I was a bit confused by your text in the Introduction that states "ARs reach the Barents and Kara Seas with increasing frequency in the months Nov, Dec, and January...". This isn't the season you focus on in the manuscript. Have there been any other studies characterizing AR events during the melt season of June-Sept?

Response: Thank you for the comment.

So far, an important feedback between sea ice and ARs that reduces the sea ice recovery in the autumn/winter months received most attention. All previous studies regarded either the autumn/winter period or yearly aggregates, while, in contrast, the current study focusses on the melt season.

We agree that citing a study focussing on a different season is not helpful. We found a very recent study that covers near future AR frequency projections. We replace the concerned sentence with the following.

Recently, Ma, Wang, Chen, *et al.* [2] reported an expected AR frequency increase for 2024 to 2064 compared to 1981 to 2021 for most of the Arctic with especially high rates over the Barents Sea.

- [2] W. Ma, H. Wang, G. Chen, *et al.*, “The role of interdecadal climate oscillations in driving arctic atmospheric river trends,” *Nature Communications*, vol. 15, no. 1, p. 2135, Mar. 8, 2024. DOI: 10.1038/s41467-024-45159-5.

Authors' Response to Reviewer 2

General Comments. What is the frequency of moist conditions during the melt season vs during winter season? You have shown, in Fig. S7, that there appears to be an increase in the number of days of moist conditions during the melt season. Fig. 2c additionally shows that there is no appreciable difference in the snowfall component of SMB over the period 1981-2022, which seems to rule out an increase in snowfall helping to offset some of the increased loss due to melt, but it doesn't necessarily address the potential for increased liquid precipitation (and therefore melt) outside of the "melt season". With an increase in the number of wintertime rain events in the Arctic (e.g., Bintanja and Andry, 2017), one might expect that an increase in moist conditions outside of the melt season could be at least partly responsible for the observed increase in mass loss.

I recognize that a thorough investigation of this is likely at least a whole study in its own right, but I think it should be possible to use your observations and data to show whether there are any appreciable changes in moist conditions during the winter season. This might also help further solidify the point you make on page 9 at line 24 ("We could not find evidence [...]") - at least, it might help confirm that the bulk of the observed increase in melt is occurring due to the changes observed during the melt season.

Response: Thank you for the comment.

The moist and dry condition classification of the submitted article explicitly focusses on the melt season, Jun–Sep, by using the melt season median moisture transport between 1981 and 2010 as reference value. Directly applying this method to the accumulation season, Oct–May, does not show a change through time with a mean of 24 days and a standard deviation of 7 days. However, it is more appropriate to, in analogy, use the accumulation season median as reference. Doing so shows that (winter-time) moist conditions become significantly more frequent with an increase of roughly 9% in the

Figure 4: Number of days with above-median moisture transport during accumulation season. The dotted line shows the original data, the solid line shows a low-pass filtered version, the dashed line shows the average for the periods 1981–2010 and 2011–2022, and the shaded area the associated 2σ -confidence intervals.

accumulation seasons between 2011 and 2022 compared to the reference period, 1981 to 2010. Fig. 4 visualizes the associated data.

This is in line with the projections of Bintanja and Andry [1]. However, we do not find that snowfall anomalies are a key factor; the snowfall increased slightly ($1.8 \pm 2.6 \text{ Gt yr}^{-1}$) in 2011 to 2022 compared to 1981 to 2010. Further, according to MAR there is still very little rainfall (0.1 Gt yr^{-1} in 2011 to 2022) in the accumulation season with no significant change ($p = 0.2$) compared to 1981 to 2010. In the model, the rainfall has a minor impact on the surface mass balance; in the accumulation season, mainly, by partly being retained in the snowpack. We note that Bintanja and Andry [1] consider the changes per season, with autumn presumably starting 1 September. We discuss the SMB changes in September in Sec. 2.2. As you mention, for stronger conclusions more research needs to be done. This is outside of the scope of the current article.

We, now, point to the projections of Bintanja and Andry [1] and add that a precipitation surplus can be expected to be rainfall. We replace the sentence:

We could not find evidence that atmospheric-river-enhanced snowfall on the Kara side plays a role; on the contrary, the albedo anomalies are more negative along the Kara coast (see supplement Fig. S8) because the moisture is precipitated as rain.

With the two sentences below (former Fig. S8, now, is Fig. S12):

Theoretical snowfall surpluses on the Kara side, that would increase the albedo and reduce melt from absorbed shortwave radiation, are not expected because the increase of moisture transport would result in rainfall [1]. We, indeed, find that the albedo anomalies are more negative along the Kara coast (see supplement Fig. S12) because the moisture is precipitated as rain.

- [1] R. Bintanja and O. Andry, “Towards a rain-dominated Arctic,” *Nature Climate Change*, vol. 7, no. 4, pp. 263–267, Apr. 2017. DOI: [10.1038/nclimate3240](https://doi.org/10.1038/nclimate3240).

Here, I think you can better motivate this study by (a) switching the order of the last two sentences ("The increased ice loss [...] of many [5]. Despite this, confidence [...]"), followed by a brief summary of some of the reasons why these projections are incomplete, and how what you have shown here points towards one area to focus on: namely, that most (all?) projections, especially at the global scale, do not adequately account for the impacts of extreme weather events.

Response: Thank you for the comment and your suggestion.

We followed your suggestion by changing the last sentences of the introduction paragraph to the following.

The increased ice loss is to produce an accelerated contribution to sea level rise, affecting the livelihoods of many [2]. Despite this, the confidence in glacier change projections is not very high [3]. These projections are based on the coarsely resolved input of General Circulation Models and, further, mostly use temperature-index based models to estimate the surface melt. Both inhibit the accurate representation of extreme weather events and the associated surface melt, respectively.

- [2] M. Oppenheimer, B. Glavovic, J. Hinkel, *et al.*, "Sea Level Rise and Implications for Low-Lying Islands, Coasts and Communities," in *The Ocean and Cryosphere in a Changing Climate: Special Report of the Intergovernmental Panel on Climate Change*, H.-O. Pörtner, D. Roberts, V. Masson-Delmotte, *et al.*, Eds., Cambridge: Cambridge University Press, 2019, pp. 321–445.
- [3] B. Marzeion, R. Hock, B. Anderson, *et al.*, "Partitioning the uncertainty of ensemble projections of global glacier mass change," *Earth's Future*, vol. 8, no. 7, e2019EF001470, 2020. DOI: 10.1029/2019EF001470.

Is this based on the RGI v6, or on the reference to Millan et al. (which used RGI v6)?

Response: Thank you for the comment.

The volume estimate is based on Millan, Mouginot, Rabatel, *et al.* [4], the total number of glaciers and the number of tidewater glaciers are based on RGI v6 [5]. We added the missing reference to RGI.

[...] it contains approximately 7600 km³[4] of ice distributed among its 479 glaciers, of which 36 terminate into the sea[5].

- [4] R. Millan, J. Mouginot, A. Rabatel, and M. Morlighem, “Ice velocity and thickness of the world’s glaciers,” *Nature Geoscience*, vol. 15, no. 2, pp. 124–129, 2 Feb. 2022. DOI: 10.1038/s41561-021-00885-z.
- [5] RGI Consortium, *Randolph Glacier Inventory - A Dataset of Global Glacier Outlines*, version 6, Boulder, Colorado USA: NSIDC: National Snow and Ice Data Center, 2017. DOI: 10.7265/N5-RGI-60.

What are the colors of the annual MB patches meant to indicate? This should be included in a legend or color bar along the side.

Response: Thank you for the comment.

The colors are meant to help identifying the mass loss of the single glaciological years and highlight the variability within the time series.

We follow your recommendation and, now, include a colorbar. The figure is reprinted, here, as Fig. 5 for convenience.

Figure 5: The mass change of the ice cap on Novaya Zemlya since 1 January, 1980, derived from the input-output (I/O) model. We include observations from GRACE/GRACE-FO[6], ICESat[7], and CryoSat-2 for which we chose arbitrary starting points such that the differences to each other are minimized but retain a 50 Gt-offset to the I/O time series to declutter the plot. The dashed lines show least-square regression results of the I/O data for the periods 1981 to 2010 and 2011 to 2022 and of the CryoSat-2 data for 2010 to 2022. The colored patches along the I/O time series quantify the prevailing negative glacier mass balance (MB) for glaciological years, i.e., starting 1 October; those are aggregated into the inset histogram.

- [6] B. Wouters, A. S. Gardner, and G. Moholdt, “Global glacier mass loss during the GRACE satellite mission (2002-2016),” *Frontiers in Earth Science*, vol. 7, 2019. DOI: 10.3389/feart.2019.00096.
- [7] G. Moholdt, B. Wouters, and A. S. Gardner, “Recent mass changes of glaciers in the Russian High Arctic,” *Geophysical Research Letters*, vol. 39, no. 10, 2012. DOI: 10.1029/2012GL051466.

Comment 4

p. 3, l. 1-3

"difference modeled the climatic surface mass balance": do you mean to say that you "difference the modeled climatic mass balance [...] and published ice discharge estimates"?

Response: Thank you for the comment.

Yes, indeed. We corrected the mistake.

Comment 5

p. 4, l. 7-8

I'm not quite sure I follow this - do you mean that in the year 2014, these fluxes (sensible heat, incoming longwave radiation, latent heat) were all lower than the 1981-2010 average, which contributed to this being a positive MB year? Or was the positive MB year in 2014 in spite of these fluxes being higher than the 1981-2010 average?

Response: Thank you for the comment.

We meant to express the former: sensible heat, incoming longwave radiation, latent heat were lower than the long-term average and shortwave was roughly equal. The sentence was not well-written. We rephrased this sentence and the previous to:

The SEB anomalies are especially pronounced for the high-melt years. Contrary, in 2014 the SEB anomalies were reversed, contributing to an overall positive mass balance for this year.

Comment 6

p. 4, l. 17

Could this [the evolution of daily surface melt above a low-pass filtered version of itself] be shown in the supplemental material?

Figure 6: Melt above its low-pass filtered version (cmp. orange highlighted areas in Fig. S6). Dotted lines show raw data, solid lines show a low-pass filtered version, dashed lines indicate the averages from 1981 to 2010 and 2011 to 2022, and the shaded areas indicate the 2σ -uncertainty intervals of the averages.

Response: Thank you for the comment.

Yes, that is possible. We added a corresponding figure to the supplement (Fig. S7), here shown as Fig. 6.

Comment 7

p. 4, l. 23

Is this difference comparing 2011-2022 with 1981-2010?

Response: Thank you for the comment.

That is correct. We added this information in the concerned sentence and the following one.

In 2011 to 2022, the absolute mass of ice melted in moist conditions increased substantially compared to 1981 to 2010 ($9 \pm 7 \text{ Gt yr}^{-1}$), while the melt in *dry conditions*, i.e., on all other melt-season days, stayed stable (see Fig. 2). Moist conditions occurred more often ($+12 \pm 9 \text{ d}$ per melt season) in 2011 to 2022, compared to 1981 to 2010.

Comment 8

p. 4, l. 27

How much snowfall was there during this year (2018)?

Response: Thank you for the comment.

In the glaciological year 2018 (Oct 2017 until Sep 2018), there were $16 \pm 2 \text{ Gt}$ snowfall. This compares to $14 \pm 2 \text{ Gt}$ in 2013. While the 2 Gt difference is not significant, it “mitigates” the total 2018 ice loss (surface processes only), such that the 2018-SMB is not significantly different from the 1981 to 2010 average ($-10 \pm 4 \text{ Gt}$ and $-3 \pm 12 \text{ Gt yr}^{-1}$, respectively).

We added the 2018 snowfall value in the manuscript.

We find that, also, in 2018, which did not qualify as high-mass-loss year because of strong snowfall (16 ± 2 Gt), there was 27 Gt melt during moist conditions, roughly equal to 2013.

Comment 9

Figure 2

Is it possible to somehow indicate the number of AR episodes here?

Response: Thank you for the comment.

This is a logical suggestion, but we did not add it for practical reason. The reason is that the number of ARs is very similar to the time ratio that moist conditions prevail. We cannot exchange the two for consistency reasons. We do already show the number of days that ARs prevail in the supplement Fig. S11.

We added a note on the link between moist conditions and ARs, and a reference to the supplement Fig. S11 to the description of Fig. 2.

Comment 10

Figure 3 caption

"The arrows [...] categories, moist and dry conditions, in shades of orange (deep and pale, respectively)". Given that the two columns of the figure already differentiate between moist and dry conditions, I'm not sure that you need to further indicate this using the arrow colors here - I think you can leave these as a single color and end the caption at "850 hPa".

Response: Thank you for the comment.

We agree and followed your suggestion. We, now, use a single color and end the caption at "850 hPa".

This is in moist easterly conditions, right? As I interpret Fig. 3, the LHF anomaly is negative on the lee slope under moist westerly conditions, and positive on the lee slope in moist easterly conditions.

Response: Thank you for the comment.

Yes, you are correct, this is in moist easterly conditions.

The concerned sentences were a confusing result from unfinished editing that slipped our attention. We corrected this issue to read:

The surface latent heat flux anomalies are negative on the lee slopes, except for the case of moist easterlies. We associate the negative anomalies with surface-melt-induced evaporative cooling. The positive latent heat anomalies on the lee-slope in moist easterly conditions are nevertheless too small an energy source to explain the observed mass loss.

Are you able to quantify the MB or melt impact of this additional SW radiation?

Response: Thank you for the comment.

In general, yes, it should be possible to estimate the MB impact with MAR by performing new sensitivity experiments (with a constant albedo). However, simulating again the full time series with MAR will take at least one month of computer time. Since this estimation is not in the main scope of the paper and in view of the requested computer resources, we prefer to not provide estimates of the impact of this additional SW radiation.

Why is the snow drift not switched on? Is there an indication that drifting snow does not play a large role in the snowpack here?

Response: Thank you for the comment.

The snow drift is negligible and, in fact, only increases the uncertainty. In favor of a simpler, more robust model, the snowdrift is not considered. We changed the sentence to read:

[...]; except the snow drift which is negligible and not switched on, here [8].

- [8] H. Gallée, G. Guyomarc'h, and E. Brun, "Impact of snow drift on the Antarctic Ice Sheet surface mass balance: Possible sensitivity to snow-surface properties," *Boundary-Layer Meteorology*, vol. 99, no. 1, pp. 1–19, Apr. 1, 2001. DOI: 10.1023/A:1018776422809.

Can you briefly state the difference between these two radiative schemes?

Response: Thank you for the comment.

The radiative schemes of ERA40 and ERA5 are not substantially different. However, there were improvements over the Greenland ice sheet.

Do you have an indication of how much this might be overestimating the mass, by undercounting the firn? If I understand this correctly, you assume that anything below the previous minimum elevation value is ice - in effect, that the previous minimum value represents ice. This is reasonable for cells at lower elevations, but what about at higher elevations? If I have misunderstood this, and "minimum value" and "preceding record" are not somehow referring to surface elevation, I am happy to be corrected here.

Response: Thank you for the comment.

We do not think that this method overestimates the mass change. We noted that our explanation did rather focus on the resulting method than the reasoning which seems confusing. To explain: We assume that the firn thickness does not decrease below an unknown minimum thickness that can vary spatially (and can, indeed, be zero). We assume this minimum thickness is reached whenever the thickness reached a new minimum. The latter assumption is justified by the long-term sustained mass loss on Novaya Zemlya. From the assumptions, it follows that the lost volume has the density of glacial ice, while the top layer of each column still is firn in any case.

We rephrase the concerned paragraph. It reads now:

We convert the volume trends and the intra-annual mass changes by a dual density model. Because Novaya Zemlya experienced sustained mass loss for a long period, we assume that a surface elevation decrease below the minimum value observed in the preceding record for each cell results from loss of glacial ice and that the associated lost volume has the density of $\rho_{ice} = 917 \text{ kg m}^{-3}$. This assumption still permits the existence of a firn layer, which has a constant thickness for each grid cell. Above the observed minimum, we assume that surface elevation changes are mainly due to the gain or loss of young firn with a density of $\rho_{firn} = 650 \text{ kg m}^{-3}$.

How have you derived u_ρ here? In the preceding paragraph, you state your values for $u_\rho^{\text{short-term}}$ and $u_\rho^{\text{long-term}}$, but do not describe how you combine these to get u_ρ .

Response: Thank you for the comment.

We did not combine the two different density uncertainty assumptions, but used the long-term value. We added the missing label and, further, clarified in the text that the use of the two uncertainty assumptions is context dependent.

Furthermore, we assume that the uncertainty of the glacierized area u_A/A is 5%, and that the uncertainty in the ice density $u_\rho^{\text{long-term}}$ is 60 kg m^{-3} over the 2010 to 2022 period and $u_\rho^{\text{short-term}} = 150 \text{ kg m}^{-3}$ for yearly changes, depending on the context [9], [10]. We assume that long-term changes of the average penetration depth of the radar signal into the firn are sufficiently acknowledged by the density uncertainty. The resulting total uncertainty for the October mass anomaly estimates that are used to derive the long-term mass balance and the yearly changes is on average 6.8 Gt. For the long-term mass balance uncertainty $u_{\dot{M}}$ these assumptions yield:

$$\begin{aligned} u_{\dot{M}} &= \sqrt{(A\rho u_h)^2 + (\dot{h}A u_\rho^{\text{long-term}})^2 + (\dot{h}\rho u_A)^2} \\ &= \sqrt{1.6 + 0.7 + 0.4} \text{ Gt yr}^{-1} = 1.6 \text{ Gt yr}^{-1} \end{aligned}$$

- [9] L. Jakob and N. Gourmelen, “Glacier mass loss between 2010 and 2020 dominated by atmospheric forcing,” *Geophysical Research Letters*, vol. 50, no. 8, e2023GL102954, 2023. DOI: 10.1029/2023GL102954.
- [10] M. Huss, “Density assumptions for converting geodetic glacier volume change to mass change,” *The Cryosphere*, vol. 7, no. 3, pp. 877–887, May 27, 2013. DOI: 10.5194/tc-7-877-2013.

Comment 17

p. 14, l. 7

It might be clearer to write this as $\Delta_{netSW} = -SWD \cdot \Delta\alpha$ - as written, it is easy to miss the \cdot , making it seem like this is $SWD - \Delta\alpha$.

Response: Thank you for your suggestion.

Yes, indeed. We followed your suggestion.

Comment 18

p. 15, ref. 5

Still "in press"?

Response: Thank you for the comment.

No. We corrected the reference.

Comment 19

p. S1, Table S2

These values are, presumably, averaged over different areas - can you include those areas here, to give an idea of the partitioning/impact on the total mass balance from each area (ice divide, lower elevations, etc)?

Response: Thank you for your suggestion.

We include the areas, now.

Comment 20

p. S4, Fig. S3

Can you show the outline of the divide area on this map?

Response: Thank you for your suggestion.

We do so, now. Additionally, following Reviewer 3's suggestion, we include the 500 m-elevation line (cmp. Comment 15).

Figure 7: CryoSat-2-observed average surface elevation change rates 2011–2022. The black, orange, and green outlines indicate the total glacierized area, the 500 m-elevation line, and the “ice divide” (in analogy with Moholdt, Wouters, and Gardner [7]) as used in Table S2.

- [7] G. Moholdt, B. Wouters, and A. S. Gardner, “Recent mass changes of glaciers in the Russian High Arctic,” *Geophysical Research Letters*, vol. 39, no. 10, 2012. DOI: 10.1029/2012GL051466.

Authors' Response to Reviewer 3

Major comments

Comment 1

The study assumed nearly constant ice discharge during 2000-2010 and 2010-2020 as quantified by the previous study. But how is this assumption plausible? For example, Melkonian et al. (2016) reported that ice flow speeds vary substantially from year to year in the region, which could potentially explain the mass loss.

Response: Thank you for the comment.

We are aware that the glacier surface velocities, and with those also the discharge, may vary in time more than in our interpolated time series. However, the gravimetry and altimetry time series we present in this article show that the increased mass loss in 2011 to 2022 can be attributed to anomalously high mass loss in the summer months of a limited number of years. Furthermore, the input-output time series which is based on the modelled surface mass balance minus the assumed mass changes from discharge agrees very favourably with these observations. Importantly, it captures the summers with anomalously high mass loss, which the SMB models shows to be caused by an unusual amount of meltwater runoff. A large-scale speed up of glaciers that would be able to explain the same amount of mass loss in such short time periods of about 3 months, with a return to its earlier speed afterwards, is from physical point of view unlikely and has to our knowledge never been observed on Novaya Zemlya.

The discharge thus plays a minor role for the current study, in which we focus on the years with large mass losses. In the case of Novaya Zemlya, the surface incurred losses are much larger than the discharge, even if the entire discharge would double for short periods. Such a doubling was reported for a single glaciers in Melkonian, Willis, Pritchard, *et al.* [1], but is unlikely to occur at an ice cap wide scale.

We conclude that, as used in the article, the assumption that the discharge does only change slowly, is permissible. We acknowledge, that this would not hold for single glaciers. We, now, include the following paragraph in Methods: Input-output model.

We acknowledge that the simplicity of the chosen discharge model neglects possible inter-glacier and temporal variability, such as seasonal summer glacier speed up [2]. This is justified by the good agreement between the model and observational data from gravitational and geodetic measurements.

- [1] A. K. Melkonian, M. J. Willis, M. E. Pritchard, and A. J. Stewart, “Recent changes in glacier velocities and thinning at Novaya Zemlya,” *Remote Sensing of Environment*, vol. 174, pp. 244–257, Mar. 1, 2016. DOI: 10.1016/j.rse.2015.11.001.
- [2] B. J. Wallis, A. E. Hogg, J. M. van Wessem, B. J. Davison, and M. R. van den Broeke, “Widespread seasonal speed-up of west Antarctic Peninsula glaciers from 2014 to 2021,” *Nature Geoscience*, vol. 16, no. 3, pp. 231–237, Mar. 2023. DOI: 10.1038/s41561-023-01131-4.

Comment 2

Somewhat related to my first point, I understand that the authors are explaining that the surface mass balance dominates the mass budget because of the similar annual specific mass balance between land- and marine-terminating glaciers. However, how reliable are the mass changes derived from altimeter for the glaciers which has a relatively narrow ice front? The authors explained that the mass change was quantified using a 500 x 500 m gridded dataset, while the glaciers have several kilometers width at the lower elevations. This limited resolution of the data may underestimate the mass change of the dynamic thinning of the marine-terminating glaciers. Maybe you could compare your altimetry-derived specific mass change with the recently published geodetic mass change in the region (for example by Sommer et al. 2022)?

Response: Thank you for the comment.

Since, as you state, the glacier fronts have a width of several kilometers, the 500 m resolution of our altimetry data is sufficient to study the mass changes at the tidewater glacier fronts.

The data density from CryoSat-2 does degrade below elevations of 100 m. We use the commonly-used *local polynomial hypsometric interpolation method* [3] to fill those voids. The concerned grid cells account for 1.4% of the total.

The current results agree with those published in Sommer, Seehaus, Glazovsky, *et al.* [4]. Sommer, Seehaus, Glazovsky, *et al.* [4] investigates changes of the ice caps in the High Russian Arctic between 2010 and 2017. For the glaciers on Novaya Zemlya, the previous study finds $-12 \pm 6 \text{ Gt yr}^{-1}$ and $-13 \pm 7 \text{ Gt yr}^{-1}$ converting geodetic measurements to mass balances using constant density models with glacial ice densities of 850 kg m^{-3} and 900 kg m^{-3} , respectively. The current data considers changes up to 2022 and, in agreement, finds a mass balance of $-12.5 \pm 1.6 \text{ Gt yr}^{-1}$. In the current study, we report surface elevation change rates of -0.77 m yr^{-1} and -0.49 m yr^{-1} along the Barents and Kara coasts, respectively (see Table S2). Sommer, Seehaus, Glazovsky, *et al.* [4] similarly finds $-0.64 \pm 0.46 \text{ m yr}^{-1}$ on average. Considering the slightly different regions and periods considered, we interpret the two datasets not to contradict each other.

We, now, include the following statement in Results: Mass loss over the past decades (cmp. response to comment 8).

The results are, further, in line with previous studies [4]–[6].

- [3] R. McNabb, C. Nuth, A. Kääb, and L. Girod, “Sensitivity of glacier volume change estimation to DEM void interpolation,” *The Cryosphere*, vol. 13, no. 3, pp. 895–910, Mar. 15, 2019. DOI: 10.5194/tc-13-895-2019.
- [4] C. Sommer, T. Seehaus, A. Glazovsky, and M. H. Braun, “Brief communication: Increased glacier mass loss in the Russian High Arctic (2010–2017),” *The Cryosphere*, vol. 16, no. 1, pp. 35–42, Jan. 6, 2022. DOI: 10.5194/tc-16-35-2022.

- [5] L. Jakob and N. Gourmelen, “Glacier mass loss between 2010 and 2020 dominated by atmospheric forcing,” *Geophysical Research Letters*, vol. 50, no. 8, e2023GL102954, 2023. DOI: 10.1029/2023GL102954.
- [6] P. Tepes, P. Nienow, and N. Gourmelen, “Accelerating ice mass loss across arctic Russia in response to atmospheric warming, sea ice decline, and atlantification of the eurasian arctic shelf seas,” *Journal of Geophysical Research: Earth Surface*, vol. 126, no. 7, e2021JF006068, 2021. DOI: 10.1029/2021JF006068.

Comment 3

Why and how do atmospheric rivers frequently occur in recent years in the region? Sea ice concentration and SST are indicated in Figure 2d but not compared with your results. They appear to be closely related to extreme melt years in 2016 and 2020. It is important to discuss how these factors are related to the occurrence of atmospheric rivers and extreme ice melting with relevant previous studies.

Response: Thank you for the comment.

The atmospheric rivers originate far from the studied region, and although interesting, it is beyond the scope of our study to resolve causes of changes in atmospheric river dynamics.

We fully agree with the reviewer that the SIC and SST may influence the energy transport to the ice cap and, now, include the following in Results: Importance & evolution of moisture imports.

We note preceding changes of sea-ice concentration (SIC) and sea surface temperature (SST), visible in Fig. 2.d. A low SIC and a high SST could facilitate that more energy arrives at the ice cap. Especially for 2016 and 2020, the SIC and SST were exceptionally low and high, respectively. We show a comparison of the skin temperature and the 2 m-air-temperature in the supplement Fig. S8.

Further comments on the main article

In the following, each comment is accompanied by a reference to the location of the article that the comment regards. The reference is in the comment's header on the right hand side. If a piece of text is referenced, we give the page number, the paragraph number, and an additional identification of the location in detail. Paragraph numbers follow the §-sign and are counted starting from the first line, even if this is the continuation of a paragraph from the preceding page.

Comment 4

P.2 §2, last sentence

Could it be more specific? For example, Melkonian et al., 2016 is rather showing a strong increase of ice speed in recent years?

Response: Thank you for the comment.

We mention the increase in discharge explicitly in the paragraph before the concerned one. For clarity, we added a note regarding the observed acceleration including references to Melkonian, Willis, Pritchard, *et al.* [1] and Carr, Bell, Killick, *et al.* [7]. The paragraph, now, ends in:

However, the discharge estimates of later studies [5], [8] do not explain the increase in mass loss despite the observed acceleration of tidewater glaciers [1], [7].

- [1] A. K. Melkonian, M. J. Willis, M. E. Pritchard, and A. J. Stewart, "Recent changes in glacier velocities and thinning at Novaya Zemlya," *Remote Sensing of Environment*, vol. 174, pp. 244–257, Mar. 1, 2016. DOI: 10.1016/j.rse.2015.11.001.
- [5] L. Jakob and N. Gourmelen, "Glacier mass loss between 2010 and 2020 dominated by atmospheric forcing," *Geophysical Research Letters*, vol. 50, no. 8, e2023GL102954, 2023. DOI: 10.1029/2023GL102954.

- [7] J. R. Carr, H. Bell, R. Killick, and T. Holt, “Exceptional retreat of Novaya Zemlya’s marine-terminating outlet glaciers between 2000 and 2013,” *The Cryosphere*, vol. 11, no. 5, pp. 2149–2174, Sep. 8, 2017. DOI: 10.5194/tc-11-2149-2017.
- [8] W. Kochtitzky, L. Copland, W. Van Wychen, *et al.*, “The unquantified mass loss of Northern Hemisphere marine-terminating glaciers from 2000–2020,” *Nature Communications*, vol. 13, no. 1, pp. 1–10, 1 Oct. 11, 2022. DOI: 10.1038/s41467-022-33231-x.

Comment 5

P.3 §1, “The yearly deficit increased to [...]”

It sounds a bit awkward choice of word.

Response: Thank you for the comment.

We replace “deficit” by “mass loss”.

Comment 6

P.3 §1, “of which only $-1.7 \pm 1.9 \text{ Gt yr}^{-1}$ [...]”

Where this number come from?

Response: Thank you for the comment.

The discharge increase is calculated from the I/O model and composes part of the difference between the mass balances of 1981 to 2010 and 2011 to 2022. For clarity, we rephrased the concerned sentence to the following.

The yearly mass loss increased to $-12.4 \pm 2.3 \text{ Gt yr}^{-1}$ in 2011 to 2022. Of the $7.1 \pm 3.3 \text{ Gt yr}^{-1}$ difference between 1981 to 2010 and 2011 to 2022, $1.7 \pm 1.9 \text{ Gt yr}^{-1}$ result from the modeled increase of ice discharge by tidewater glaciers and $5.5 \pm 2.7 \text{ Gt yr}^{-1}$ are lost via surface processes, mainly meltwater runoff.

Comment 7**P.3 §1, “The remainder of $-5.5 \pm 2.7 \text{ Gt yr}^{-1}$ [...]”**

I could not possible to follow how you calculate this number.

Response: Thank you for the comment.

See the answer to previous Comment 6.

Comment 8**P.3 §2, 2nd sentence**

You could also compare your result with more recent geodetic mass change (Sommer et al., 2022). It is also important to compare the number for each basin. I concern the altimetry derived mass loss underestimate the mass loss for the calving glaciers (see my major comment).

Response: Thank you for the comment.

See answer to Comment 2, for more detail.

We compared our geodetic mass balance results to that of gravimetry derived and modeled mass balances and find that all agree. We, now, also compare the current results to Sommer, Seehaus, Glazovsky, *et al.* [4].

- [4] C. Sommer, T. Seehaus, A. Glazovsky, and M. H. Braun, “Brief communication: Increased glacier mass loss in the Russian High Arctic (2010–2017),” *The Cryosphere*, vol. 16, no. 1, pp. 35–42, Jan. 6, 2022. DOI: 10.5194/tc-16-35-2022.

Comment 9**P.4 Section 2.3, 3rd last sentence, “Especially, [...]”**

How those years relate to sea ice and SST in the region as well as large-scale climate variabilities? It would be nice if the authors discuss it somewhere in the manuscript.

Response: Thank you for the comment.

We, now, include a discussion of the sea ice and SST evolution. See answer to Comment 3.

Comment 10

Figure 2, Panel (d)

Looks like this panel was never cited in the manuscript, but it would be interesting to compare with your results.

Response: Thank you for the comment.

Agreed. We, now, do so. See answer to Comment 3.

Comment 11

Figure 3

I found this plots are hard to follow because lots of information packed in. You should think of labeling panels with a, b, etc and cite specific panel in the main text.

Response: Thank you for your suggestion.

We followed your suggestion. The figure has been reprinted as Fig. 1 above, in the context of Reviewer 1's comments. The figure legend, now, states the following:

(a)-(b) Spatial distributions of melt, (c)-(d) sensible heat flux (SHF), (e)-(f) downward longwave radiation (LWD), (g)-(h) latent heat flux (LHF), and (i)-(j) downward shortwave radiation (SWD) anomalies, stippled where insignificant ($p \geq 0.05$). The vertical columns distinguish moist or dry conditions (see Results: Importance & evolution of moisture imports). Of each pair, the left image illustrates the average over days with westerly winds, the right one averages days with easterly winds. The orange arrows in the first row, (a)-(b), indicate the moisture-transport-weighted wind direction azimuth bearing at 850 hPa.

Comment 12

P.7 §2, entire paragraph

This paragraph was a bit hard to follow because no panel label in the Figure 3.

Response: Thank you for the comment.

We labeled the plot and, now, use the labels to refer to the panels.

Comment 13

P.10 Section 4.2, last sentence

How the model performs over Novaya Zem[[]]ya but not over Greenland? You should rather evaluate your model input and output with ground truth obtained at Novaya Zemlya.

Response: Thank you for the comment.

Comparing to ground truth data obtained at Novaya Zemlya would be ideal, but unfortunately there are no field data available that would enable such an evaluation. We compare the model to two independent observational data sources, namely GRACE/GRACE-FO and CryoSat-2. Moreover, Maure, Kittel, Lambin, *et al.* [9] recently inspected the MAR model output for this region. We make this explicit by adding “,including Novaya Zemlya” when citing the regarded source. Finally, the climate of Greenland and Novaya Zemlya are similar and a good performance over Greenland indicates that we can expect a good performance over Novaya Zemlya.

- [9] D. Maure, C. Kittel, C. Lambin, A. Delhasse, and X. Fettweis, “Spatially heterogeneous effect of climate warming on the arctic land ice,” *The Cryosphere*, vol. 17, no. 11, pp. 4645–4659, Nov. 6, 2023. DOI: 10.5194/tc-17-4645-2023.

Comment 14P.11 §1, “we interpolate the discharge [...]”

How much this assumption plausible?

Response: Thank you for the comment.

See answer to Comment 1.

Comments on the supplement

Comment 15

Fig. S3

You could indicate the 500 m elevation contour to highlight the ice front and ice divide regions?

Response: Thank you for your suggestion.

We followed your suggestion. See Fig. 7, previously printed in the context Reviewer 2's Comment 20.

Comment 16

Fig. S4

Could you specify which marker represents which year by color or text especially for 2013, 2016, 2020 and 2022?

Response: Thank you for your suggestion.

We followed it and, now, highlight all years that are mentioned in the main article, namely 2013, 2014, 2016, 2018, 2020, and 2022. We reprint the figure, here, as Fig. 8.

Comment 17

Fig. S5

What is the shaded area means?

Response: Thank you for the comment.

The shaded areas indicate the 2σ -confidence interval of the averages, shown as dashed lines.

We, now, also clarify this in the figure's legend by adding:

Figure 8: Comparison of yearly area-specific mass changes of marine- and land-terminating glaciers. Those years that were mentioned in the main article can be identified using the legend; other years are colored in blue.

The dashed lines and shaded areas show the averages and their 2σ -confidence intervals for the periods 1981 to 2010 and 2011 to 2022.

References

- [1] A. K. Melkonian, M. J. Willis, M. E. Pritchard, and A. J. Stewart, “Recent changes in glacier velocities and thinning at Novaya Zemlya,” *Remote Sensing of Environment*, vol. 174, pp. 244–257, Mar. 1, 2016. DOI: 10.1016/j.rse.2015.11.001.
- [2] B. J. Wallis, A. E. Hogg, J. M. van Wessem, B. J. Davison, and M. R. van den Broeke, “Widespread seasonal speed-up of west Antarctic Peninsula glaciers from 2014 to 2021,” *Nature Geoscience*, vol. 16, no. 3, pp. 231–237, Mar. 2023. DOI: 10.1038/s41561-023-01131-4.
- [3] R. McNabb, C. Nuth, A. Kääh, and L. Girod, “Sensitivity of glacier volume change estimation to DEM void interpolation,” *The Cryosphere*, vol. 13, no. 3, pp. 895–910, Mar. 15, 2019. DOI: 10.5194/tc-13-895-2019.
- [4] C. Sommer, T. Seehaus, A. Glazovsky, and M. H. Braun, “Brief communication: Increased glacier mass loss in the Russian High Arctic (2010–2017),” *The Cryosphere*, vol. 16, no. 1, pp. 35–42, Jan. 6, 2022. DOI: 10.5194/tc-16-35-2022.
- [5] L. Jakob and N. Gourmelen, “Glacier mass loss between 2010 and 2020 dominated by atmospheric forcing,” *Geophysical Research Letters*, vol. 50, no. 8, e2023GL102954, 2023. DOI: 10.1029/2023GL102954.
- [6] P. Tepes, P. Nienow, and N. Gourmelen, “Accelerating ice mass loss across arctic Russia in response to atmospheric warming, sea ice decline, and atlantification of the eurasian arctic shelf seas,” *Journal of Geophysical Research: Earth Surface*, vol. 126, no. 7, e2021JF006068, 2021. DOI: 10.1029/2021JF006068.
- [7] J. R. Carr, H. Bell, R. Killick, and T. Holt, “Exceptional retreat of Novaya Zemlya’s marine-terminating outlet glaciers between 2000 and 2013,” *The Cryosphere*, vol. 11, no. 5, pp. 2149–2174, Sep. 8, 2017. DOI: 10.5194/tc-11-2149-2017.
- [8] W. Kochtitzky, L. Copland, W. Van Wychen, *et al.*, “The unquantified mass loss of Northern Hemisphere marine-terminating glaciers from 2000–2020,” *Nature*

Communications, vol. 13, no. 1, pp. 1–10, 1 Oct. 11, 2022. DOI: 10.1038/s41467-022-33231-x.

- [9] D. Maure, C. Kittel, C. Lambin, A. Delhasse, and X. Fettweis, “Spatially heterogeneous effect of climate warming on the arctic land ice,” *The Cryosphere*, vol. 17, no. 11, pp. 4645–4659, Nov. 6, 2023. DOI: 10.5194/tc-17-4645-2023.

REVIEWERS' COMMENTS

Reviewer #1 (Remarks to the Author):

Though it would be helpful to have a more definitive understanding of what AR characteristics lead to foehn events, given limitations of the current methodology, this is left for future work. Aside from this piece, the authors have sufficiently addressed any previous comments and I recommend publication.

Reviewer #2 (Remarks to the Author):

I have read the response and reviewed the latexdiff document, and I feel that my previous comments have been satisfactorily addressed. I thank the authors for their careful response and revision, and congratulate them on an interesting and well-written study that highlights the importance of extreme weather events on glacier mass balance, which could have substantial implications for projections of future glacier mass loss.

Reviewer #3 (Remarks to the Author):

Thank you for your responses regarding my concerns. I'm convinced by your replies.

They report a new mechanism that the recently increased ice mass loss could be explained by foehn winds accompanied by extreme energy imports from atmospheric rivers, which were previously believed to be caused by dynamic calving glacier behaviours. I believe the study has great potential to be published in a high-impact journal and attract significant interest from researchers across a wide range of research fields.